# Self-organizing glycolytic waves tune cellular metabolic states and fuel cancer progression

Huiwang Zhan [1] ✉, Dhiman Sankar Pal [1], Jane Borleis[1], Yu Deng [1], Yu Long[1], Chris Janetopoulos [1,2] ✉, Chuan-Hsiang Huang [1,3] ✉ & Peter N. Devreotes [1] ✉

Although glycolysis is traditionally considered a cytosolic reaction, here we show that glycolytic enzymes propagate as self-organized waves on the membrane/cortex of human cells. Altering these waves led to corresponding changes in glycolytic activity, ATP production, and dynamic cell behaviors, impacting energy-intensive processes such as macropinocytosis and protein synthesis. Mitochondria were absent from the waves, and inhibiting oxidative phosphorylation (OXPHOS) had minimal effect on ATP levels or cellular dynamics. Synthetic membrane recruitment of individual glycolytic enzymes increased cell motility and co-recruited additional enzymes, suggesting assembly of glycolytic multi-enzyme complexes in the waves. Remarkably, wave activity and glycolytic ATP levels increased in parallel across human mammary epithelial and other cancer cell lines with higher metastatic potential. Cells with stronger wave activity relied more on glycolysis than on OXPHOS for ATP. These results reveal a distinct subcellular compartment for enriched local glycolysis at the cell periphery and suggest a mechanism that coordinates energy production with cellular state, potentially explaining the Warburg effect.

Classical biochemistry holds that enzymes of the glycolytic pathway reside in the cytosol and that glycolysis takes place within the cytoplasm. Consistently, the enzymes separate into the supernatant of fractionated cells; indeed the release of soluble aldolase is often considered as a marker for cell lysis[1]. Furthermore, glycolytic activity, as measured by the conversion of glucose to pyruvate, can be reconstituted in vitro by mixing purified enzymes and substrates in appropriate concentrations in solution[2]. Inside cells, pyruvate enters mitochondria from the cytosol and initiates oxidative phosphorylation. Although oxidative phosphorylation yields more ATP per glucose molecule than glycolysis, glycolysis can generate ATP at a much faster rate[3]. Cells that need rapid ATP production can leverage glycolysis for this reason. Moreover, research from the 1920s has shown that cancer cells, which require a faster ATP source compared to their healthy counterparts due to increased activity, often shift towards a greater reliance on glycolysis, a phenomenon known as the Warburg effect[3–5]. However, the mechanism by which cancer cells shift their energy reliance to glycolysis remains unclear.

Energy-intensive dynamic cellular activities such as phagocytosis, cytokinesis, and cell migration require cell shape changes induced by signal transduction and acto-myosin cytoskeletal activities, which result in various membrane protrusions and deformations[6–13]. Emerging studies in different types of cells have unveiled fascinating links between acto-myosin dynamics at the cell cortex adjacent to the plasma membrane and pathological conditions such as cancer progression[14,15]. It is traditionally thought that ATP from glycolysis is generated in the cytosol and then diffuses to the surface of the cell to fuel these rapid local dynamic activities[16,17]. We hypothesized that glycolysis might be localized directly to the plasma membrane, providing a spatial and temporal mechanism to accelerate local ATP

[1]Department of Cell Biology and Center for Cell Dynamics, School of Medicine, Johns Hopkins University, Baltimore, MD, USA. [2]Total Experience Learning, Alvernia University, Reading, PA, USA. [3]Department of Pathology, School of Medicine, Johns Hopkins University, Baltimore, MD, USA. ✉ e-mail: davidzhan@jhmi.edu; cjaneto1@jh.edu; chuang29@jhmi.edu; pnd@jhmi.edu

production. Such regulation would be pivotal to our understanding of ATP generation, as it could reveal novel insights into the regulation of glycolysis, particularly how its activity is altered in disease states and contribute to cancer progression.

These considerations prompted us to investigate the sub-cellular localization of glycolytic enzymes in normal human and cancer cells. Surprisingly, we found that all the glycolytic enzymes we tested are enriched in the self-organized traveling waves in the plasma membrane/cortex. Increasing wave activity results in enzyme recruitment to the membrane while abolishing the waves returns the enzymes to the cytosol. Consistently, we further demonstrated that glycolytic ATP production is strongly correlated with the augmentation or abrogation of these waves, with ATP produced from these glycolytic waves making up about 33% of the overall ATP generated by glycolysis. Remarkably, recruiting a single glycolytic enzyme to the cell membrane induces epithelial cell spreading and accelerates neutrophil migration, while inhibiting glycolytic waves silences energy intensive processes such as migration, macropinocytosis, and protein synthesis. Furthermore, membrane-recruitment of phosphofructokinase causes a co-recruitment of aldolase, suggesting that these glycolytic enzymes may form a complex in the plasma membrane. Notably, there is a progressive increase in wave frequency accompanied by a rise in ATP levels across a series of cells from various tissues with increasing metastatic indices. This suggests that the greater abundance of glycolytic waves in cancer cells may underlie their increased reliance on glycolysis, a hallmark of the Warburg effect.

## Results

### Glycolytic enzymes are enriched at Lifeact labeled waves and protrusions

To visualize the localization of glycolytic enzymes, we first introduced GFP-tagged aldolase into the mammary epithelial cancer cell MCF-10A M3. We captured images of the cells' basal surfaces and made a fascinating observation: while the aldolase was distributed within the cytosol as expected, a significant fraction was found to be associated with dynamic waves moving across the basal surface of the cell (Fig. 1a, Supplementary Fig. 1a, c, and Supplementary Movie 1). Line kymographs drawn through several different planes exhibited the dynamic spatial and temporal characteristics of aldolase localization in the propagating waves (Fig. 1b and Supplementary Fig. 1b, d).

In the same cells, mCherry was used as a control and was evenly spread throughout the cytosol. Merged images revealed a clear enrichment of aldolase in the waves compared to the cytosolic mCherry signal (Fig. 1a and Supplementary Fig. 1c). To quantitatively assess the enrichment in the waves, we normalized the aldolase-GFP signal to the cytosolic mCherry signal, revealing a significantly heightened signal within the waves (Fig. 1a). Scans across the wave area of the cell showed no increase in the cytosolic mCherry control within the waves but a substantial increase for aldolase, with a ratio value ranging from 3 to 10-fold compared to other cytosolic regions (Supplementary Fig. 1e, f).

We performed the analogous analyses for the distribution of Lifeact, a marker for newly formed branched F-actin, which is also localized to propagating waves on the basal surface of cells (Fig. 1c and Supplementary Movie 2). Merged images of Lifeact and cytosolic GFP and the ratio of the two signals further supported the enhancement of Lifeact in the wave region. Line kymographs also showed the dynamic nature of the F-actin waves, absent in the GFP control (Fig. 1d). Line scans indicated a 3-fold increase in Lifeact within the waves compared to non-wave cytosolic regions (Supplementary Fig. 1g).

Interestingly, when aldolase-GFP and Lifeact-RFP were co-expressed, both markers were enriched in traveling waves and protrusions that emerged when a wave reached the cell's edge (Fig. 1e, Supplementary Movie 3). Examination of merged images revealed that the distribution of aldolase was more diffuse compared to that of Lifeact (Fig. 1e). Line kymographs illustrated similar spatial-temporal patterns of aldolase and Lifeact in the propagating waves and protrusions (Fig. 1f). Line scans across the waves confirmed that the aldolase distribution was slightly wider than that of F-actin (Supplementary Fig. 1h). We quantified the properties of these waves and found that the parameters of aldolase-GFP waves (such as wave length, band width, velocity, and duration) are highly correlated with those of Lifeact labelled actin waves that we had quantified in our previous study[14] (Supplementary Fig. 2), indicating that the aldolase waves may be coupled with actin waves.

The presence of aldolase in waves prompted further investigation into the cellular localization of other glycolytic enzymes (Fig. 2a). Of the eight remaining enzymes in the glycolysis cascade, five, namely hexokinase (HK), phosphofructokinase (PFK), glyceraldehyde 3-phosphate dehydrogenase (GAPDH), enolase (ENO), and pyruvate kinase (PK), could be tagged with GFP or RFP for visualization and successfully expressed in the epithelial cells. All five additional glycolytic enzymes tested exhibited enrichment within F-actin waves and protrusions (Fig. 2b–k and Supplementary Movies 4–8). Co-expressing each glycolytic enzyme with LifeAct, labeled with an appropriate complementary color, enabled simultaneous imaging, highlighting the dynamic localization patterns in waves and protrusions. The temporal and spatial patterns of glycolytic enzyme localization showed high coordination with waves of actin polymerization, evident from color-coded temporal overlay images (Fig. 2c, e, g, i, k). Scans through the waves showed that, as found for aldolase, the other glycolytic enzymes display more diffusive patterns than LifeAct (Fig. 2l).

To rule out the possibility that glycolytic waves are artifacts of GFP-tagged proteins, we performed immunofluorescence (IF) staining of endogenous glycolytic enzymes in fixed cells. To ensure that fixation did not alter the wave patterns, we recorded time-lapse movies of cells co-expressing PFK-GFP and LifeAct-RFP before and after treatment with 4% paraformaldehyde (PFA). As shown in Fig. 3a and Supplementary Fig. 3, the cell exhibited multiple propagating waves of PFK-GFP and LifeAct-RFP prior to fixation. Notably, upon addition of 4% PFA, wave propagation ceased, but the wave patterns at the moment of fixation were maintained. These results indicate that fixation effectively preserves the wave structure.

We then performed IF staining of various glycolytic enzymes in fixed cells transfected with LifeAct-RFP, which served as a marker for the waves. While IF staining for GFP showed no signal (Fig. 3b), IF staining for GAPDH, aldolase, enolase-1, PFK, and HK-1 revealed distinct wave-like patterns that colocalized with LifeAct-RFP (Fig. 3c–k). LifeAct-RFP expression is heterogeneous. In some cells lacking LifeAct-RFP such as shown in Fig. 3j, wave patterns of endogenous PFK remained clearly visible, indicating that the waves were not induced by the exogenous protein expression. Together, these results demonstrate that endogenous glycolytic enzymes localize to propagating waves.

While not all glycolytic enzymes were expressed in this study, our results suggest a trend of enrichment for the entire glycolysis cascade within actin waves that generate ruffles in cells[6,7,14,18]. The dynamic clustering of enzymes within these waves is anticipated to significantly elevate concentrations, thereby substantially augmenting the glycolysis rate. This observation implies that glycolytic enzymes could play a role in actin-based structures and the generation of cellular protrusions.

### Perturbations that alter waves cause parallel changes in glycolysis and ATP

To further investigate the correlation between the glycolytic enzymes and actin waves, we examined the activities of glycolytic waves under perturbations that are known to change the wave dynamics. It was previously shown that addition of EGF and insulin increases the Ras/PI3K and actin wave activity in these cells within

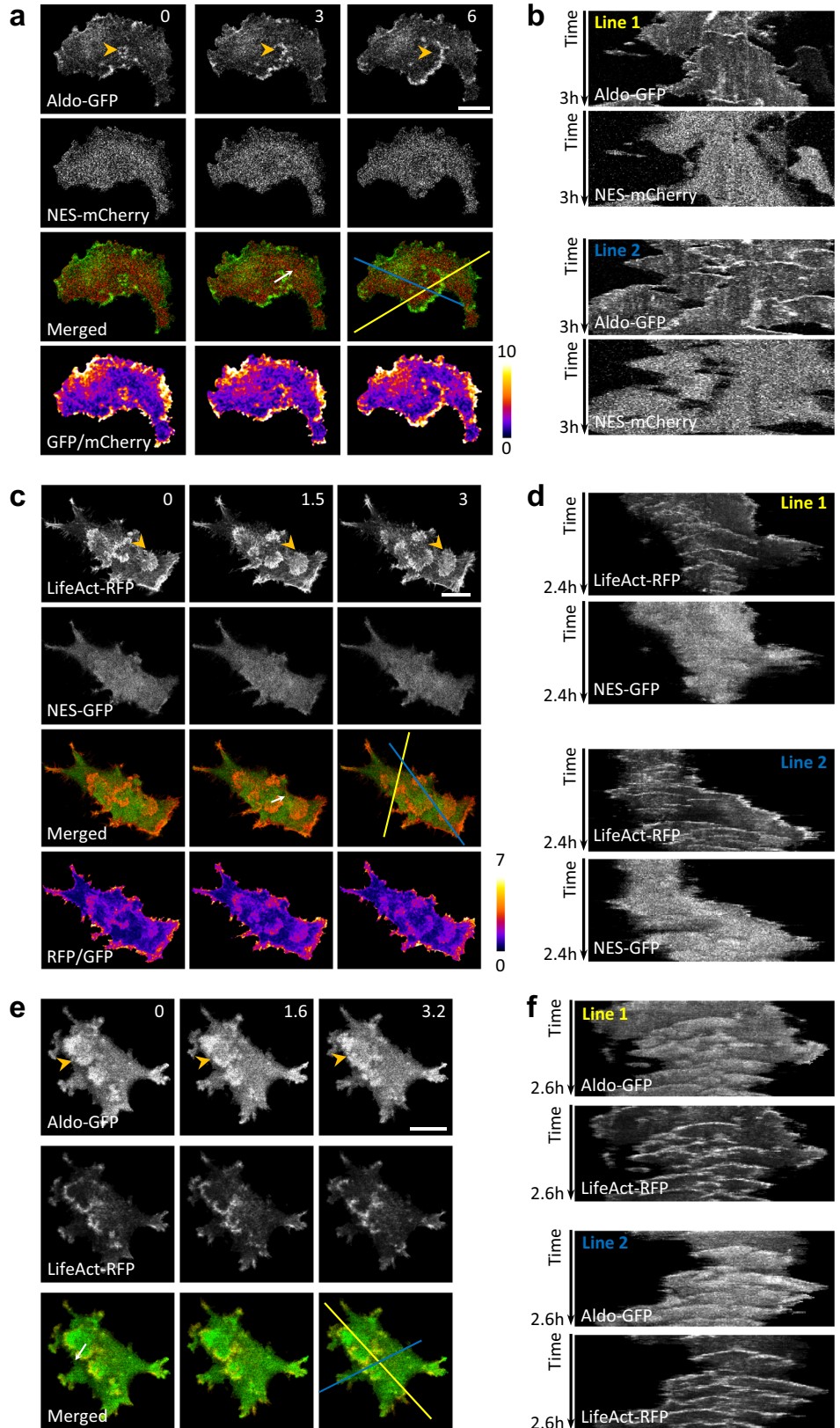

minutes and this effect persists for hours[14]. We monitored GFP tagged aldolase and PFK in the ventral surface of the MCF-10A M3 cells before and after stimulation with EGF and Insulin. While aldolase waves were present prior to the addition of EGF, the number of waves dramatically increased with the addition of EGF and Insulin (Fig. 4a and Supplementary Movie 9). The response was variable among individual cells (Fig. 4b), but on average there was a ~2.5-fold increase in wave activities which plateaued within 30 min and was maintained for the time imaged (Fig. 4c). Similar effects were observed for PFK (Fig. 4d–f and Supplementary Movie 9).

As suggested earlier, the enrichment of the glycolytic enzymes in the propagating waves may concentrate the enzymes and thus

**Fig. 1 | Enrichment of aldolase in actin waves. a** Time-lapse confocal images of the basal surface of an MCF-10A M3 cell expressing aldolase-GFP and NES-mCherry (see also Supplementary Movie 1). A zoomed-in view of the wave is shown in Supplementary Fig. 1A, and another example cell is shown in Supplementary Fig. 1c. **b** Kymographs of fluorescence intensity along the two lines in (a) over time. A zoomed-in view of the kymographs is shown in Supplementary Fig. 1b. **c** Time-lapse confocal images of the basal surface of a cell expressing LifeAct-RFP and NES-GFP (also see Supplementary Movie 2). **d** Kymographs along the two lines in (c) over time. **e** Time-lapse confocal images of the basal surface of a cell expressing aldolase-GFP and LifeAct-RFP (also see Supplementary Movie 3). **f** Kymographs along the two lines in (e) over time. The orange arrowheads in (**a, c, e**) indicate expanding waves propagating across the basal surface of the cell. The scan of fluorescence intensity across the white arrows in (**a, c, e**) are shown Supplementary Fig. 1f–h, respectively. Scales bars are 20 μm and the unit of time stamp is min throughout all figures unless otherwise indicated. Images shown in each panel represent a typical example of cells from $N \geq 3$ independent experiments respectively.

enhance the overall glycolytic activity. As a reflection of the glycolytic activity, we utilized the biosensor iATP (mRuby-iATPSnFR)[19] to measure the intracellular level of ATP. The ratio of cpGFP to mRuby is correlated with the relative level of ATP (Supplementary Fig. 4a). First, we examined the response of this biosensor to a cocktail of glycolysis inhibitors, 2-Deoxy-D-Glucose (2-DDG) and 3-BromoPyruvic Acid (3-BPA) designated DB, and to a cocktail of oxidative phosphorylation (OXPHOS) inhibitors, Oligomycin, Antimycin, and Rotenone designated OAR (Supplementary Fig. 4b). When DB was added, there was a major drop of signal in cpGFP but little change in mRuby (Supplementary Fig. 4c and Supplementary Movie 10). The ratio of cpGFP to mRuby decreased ~70% in less than 10 min after addition of DB (Supplementary Fig. 4d). Later application of OAR did not further reduce ATP (Supplementary Fig. 4d and Supplementary Movie 10). When OAR was applied first, the ratio of cpGFP to mRuby was reduced by less than 10%, while further application of DB reduced the ratio by ~60% (Supplementary Fig. 4e). These experiments suggest that in MCF-10A M3 cells, ATP is largely produced from glycolysis rather than from OXPHOS, which is consistent with previously published results[20]. Mitotracker showed that mitochondria were not localized in these waves and protrusions (Supplementary Fig. 5).

We used the ATP biosensor to measure the glycolytic activity in response to the EGF and insulin. To focus on the ATP change from glycolysis, we pretreated the cells in OAR to eliminate the ATP contribution from OXPHOS. When EGF and insulin were added, there was a burst of ATP production (Fig. 4g). The cpGFP/mRuby ratio showed a ~20% increase within 30 min (Fig. 4h and Supplementary Fig. 6a). This timing was consistent with the increase in aldolase and PFK associated wave activities as indicated in Fig. 4a–f. This finding suggests that the induced recruitment of the glycolytic enzymes, concentrating them into the waves, may account for the increase in glycolytic activity and ATP production.

We further examined the correlation of the ATP level and the wave activity by blocking the cortical association of the glycolytic enzymes. Latrunculin A (LatA) dramatically reduced the actin waves and resulted in a ~25% drop in the basal cpGFP/mRuby ratio (Fig. 4i–k and Supplementary Fig. 6b). Subsequent addition of EGF and insulin did not increase the cpGFP/mRuby ratio (Fig. 4i, l and Supplementary Fig. 6c). Under this condition, aldolase and PFK did not appear to redistribute to the plasma membrane (Fig. 4m). LatA did not block the activation of PI3K by EGF and insulin as indicated by membrane recruitment of PH-AKT, biosensor of the bioproduct of PI3K activation (Fig. 4m), suggesting that PI3K activation is not sufficient for enhancing glycolysis (Fig. 4i, l). However, upon the inhibition of PI3K by its inhibitor LY294002 (LY), the cpGFP/mRuby ratio dropped by ~25% (Fig. 4n, p). The decrease in glycolytic activity by PI3K inhibition was confirmed with pyruvate[21] and NADH/NAD+ biosensors[22] (Fig. 4o, q, Supplementary Fig. 6d–h, and Supplementary Movies 11, 12). These results are quantified in Fig. 4p, q and Supplementary Fig. 6f. Notably, there was also a dramatic drop in the wave activities of LifeAct and aldolase upon PI3K inhibition (Fig. 4r and Supplementary Movie 11). These results collectively suggest that the enrichment of glycolytic enzymes in the propagating waves leads to increased glycolytic activities and higher glycolytic ATP production.

## Relationship between glycolytic wave activities and cell motility and dynamics

To investigate the relationship between waves of glycolytic enzymes and cell behavior, we altered the wave activities with synthetic biology and drug treatment. We previously showed that abruptly lowering plasma membrane PI(4,5)P2 by recruitment of Inp54p using chemically induced dimerization (CID) (Supplementary Fig. 7a and Fig. 5a) initiates coordinated signaling and actin waves and accompanied protrusive activities in initially quiescent MCF-10A cells. Lowering PI(4,5)P2 is demonstrated to lower the threshold of the excitable biochemical network that underlies the wave activities[14]. We wondered whether aldolase would be enriched in these de novo actin waves acutely triggered by lowering PI(4,5)P2. As shown in Fig. 5a, b, upon PI(4,5)P2 reduction, cytosolic aldolase was enriched in these increased propagating waves and protrusions, which spiraled around the cell perimeter (Fig. 5c, d, Supplementary Fig. 7b, and Supplementary Movie 13). This observation is similar to the EGF and insulin induced increase in waves, protrusions, and associated aldolase and PFK. Thus, the propensity of the glycolytic enzymes to associate with the actin waves, and increase ATP production, is strongly correlated with the overall excitability of the signaling and cytoskeletal networks and the generation of F-actin-based protrusions and cellular dynamical activities.

We thus questioned whether the acute synthetic recruitment of glycolytic enzymes to the plasma membrane would facilitate the wave formation and thus accelerate glycolysis and alter cell dynamics. To explore this, we developed CID and optogenetic systems to rapidly recruit glycolytic enzymes to the membrane and assess the resulting changes in cell behaviors. Surprisingly, in MCF-10A M3 cells, recruitment of PFK to the plasma membrane by CID triggered cell spreading and the appearance of dynamic actin patches (Fig. 5e–g and Supplementary Movie 14). To demonstrate that this effect was not limited to one enzyme in one cell type, we created a light-inducible recruitment system for aldolase in neutrophil-like HL-60 cells (Supplementary Fig. 8a). Quiescent HL-60 cells became polarized and highly motile after the recruitment of aldolase to the membrane (Fig. 5h and Supplementary Movie 15). The recruited aldolase was initially uniformly localized and became enriched in the LifeAct labeled protrusions as the cell became polarized and started to migrate persistently (Fig. 5i, h and Supplementary Movie 15). As shown in Fig. 5j, k, migration speed, cell spreading, and polarity increased with aldolase recruitment. These changes were not detected in the non-recruitment controls (Supplementary Fig. 8b, c). Together, these results showed the surprising observation that relocalization of a single glycolytic enzyme from the cytosol to the plasma membrane can dramatically alter cell dynamic morphology and increase cell migration.

We next explored the effect of inhibition of metabolic pathways on cell migration. As demonstrated above, glycolysis inhibitors abolished the majority of ATP production in the cell while OXPHOS had a minor effect (Supplementary Fig. 4). Accordingly, we analyzed the effects of inhibitors of glycolysis and OXPHOS on cell migration. Similar to the drop in cpGFP/mRuby ratio of iATP, the actin wave activities (Supplementary Fig. 9a, b), dynamic morphological changes (Supplementary Fig. 9c–g) and cell migration tracks (Supplementary Fig. 9h) all dramatically decreased upon the inhibition of glycolysis, while further addition of OXPHOS inhibitors did not cause further

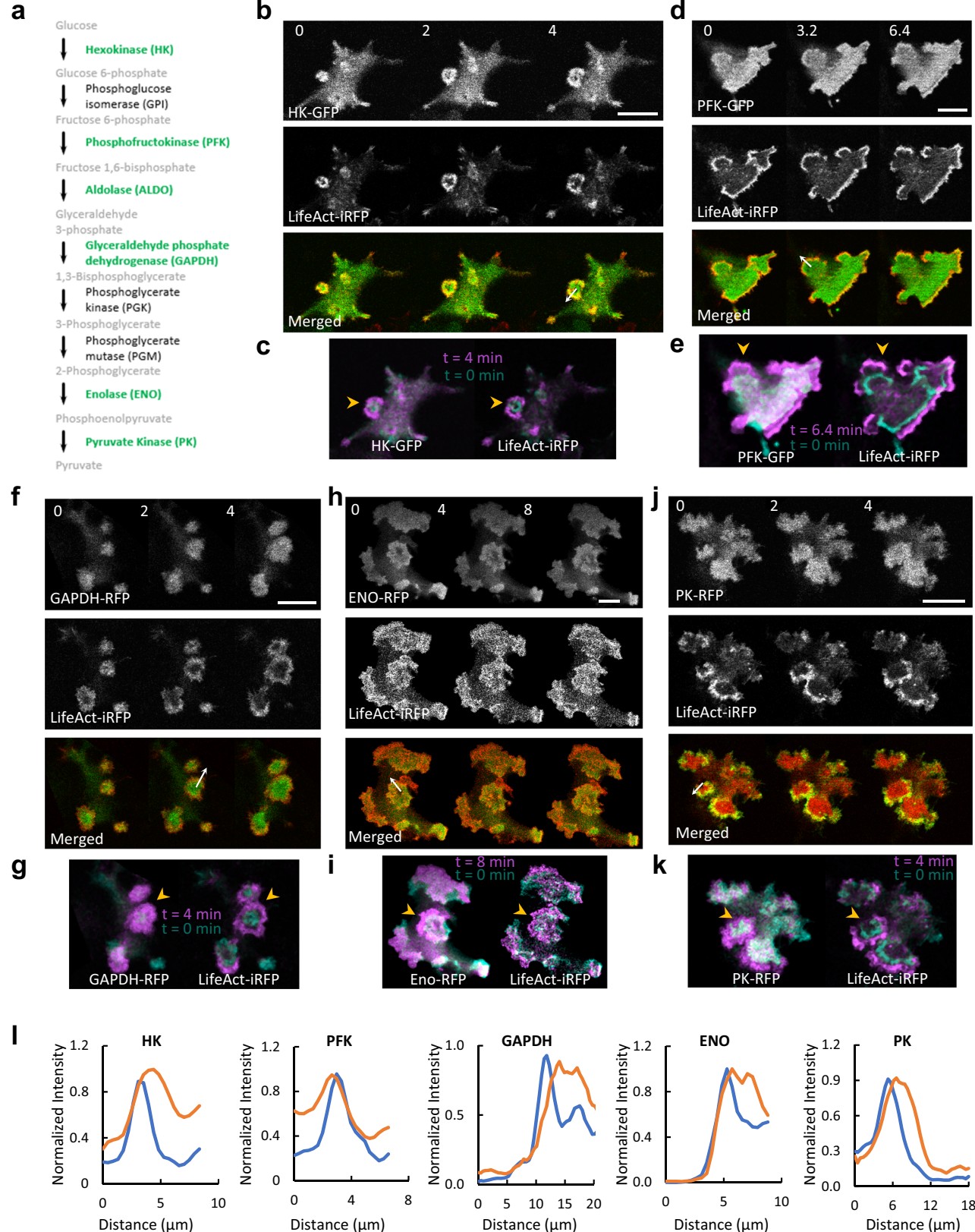

**Fig. 2 | Enrichment of additional glycolytic enzymes in actin waves. a** The gly-colytic pathway. Enzymes that could be fluorescently tagged and imaged in this study are shown in green. **b**–**k** Colocalization of LifeAct and glycolytic enzymes in waves. Time-lapse confocal images of the basal surface of MCF-10A M3 cells expressing LifeAct-iRFP670 and HK-GFP (**b**), PFK-GFP (**d**), GAPDH-RFP (**f**), Enolase-RFP (**h**), and PK-RFP (**j**) are shown (also see Supplementary Movies 4–8). Color-coded overlays show the progression of waves over time (**c**, **e**, **g**, **i**, **k**). The orange arrowheads in (**c**, **e**, **g**, **i,k**) indicate expanding waves propagating across the basal surface of the cell. Full length proteins are used except for HK, in which the first 21 a.a. of the N-terminus were truncated[69]. These images shown in (**b**–**k**) represent a typical example of cells from N ≥ 3 independent experiments respectively. **l** Normalized intensity of LifeAct-iRFP (blue) and FP-tagged glycolytic enzymes (orange) across the white arrow in (**b**, **d**, **f**, **h**, **j**).

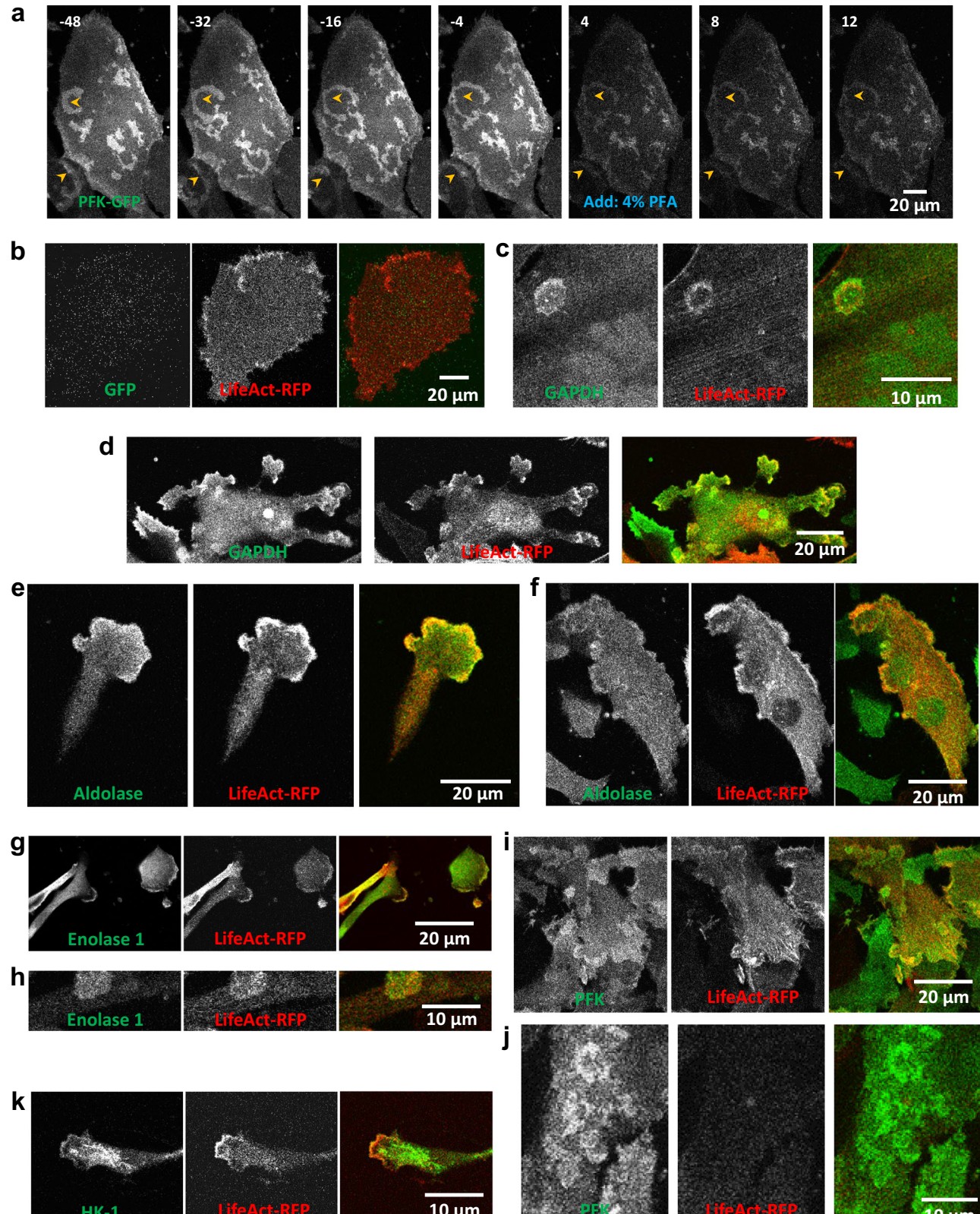

**Fig. 3 | Endogenous glycolytic enzymes are also enriched in LifeAct labelled waves. a** Time-lapse confocal images showing the basal surface of MCF-10A-M3 cells expressing LifeAct-RFP (shown in Supplementary Fig. 12) and PFK-GFP before and after fixation by 4% PFA (added at time 0). Yellow arrow heads indicate examples of waves. Time stamp is minute. The decrease in fluorescence intensity was due to partial dissipation of GFP-tagged proteins caused by cell permeabilization. Fluorescence signals after fixation were multiplied by 2 for better

presentation. This shown image represents a typical example of cells from $N \geq 3$ independent experiments. **b**–**k** Confocal images showing the basal surface of 4% PFA fixed MCF-10A-M3 cells expressing LifeAct-RFP and stained with antibodies against GFP (**b**), GAPDH (**c, d**), Aldolase (**e, f**), Enolase 1 (**g, h**), PFK (**i, j**), and HK-1 (**k**). Channels of antibody staining, LifeAct, and merged are shown. The units of the scale bar are indicated in each image. Each image shown in (**b**-**k**) represents a typical example of cells from $N \geq 3$ independent experiments.

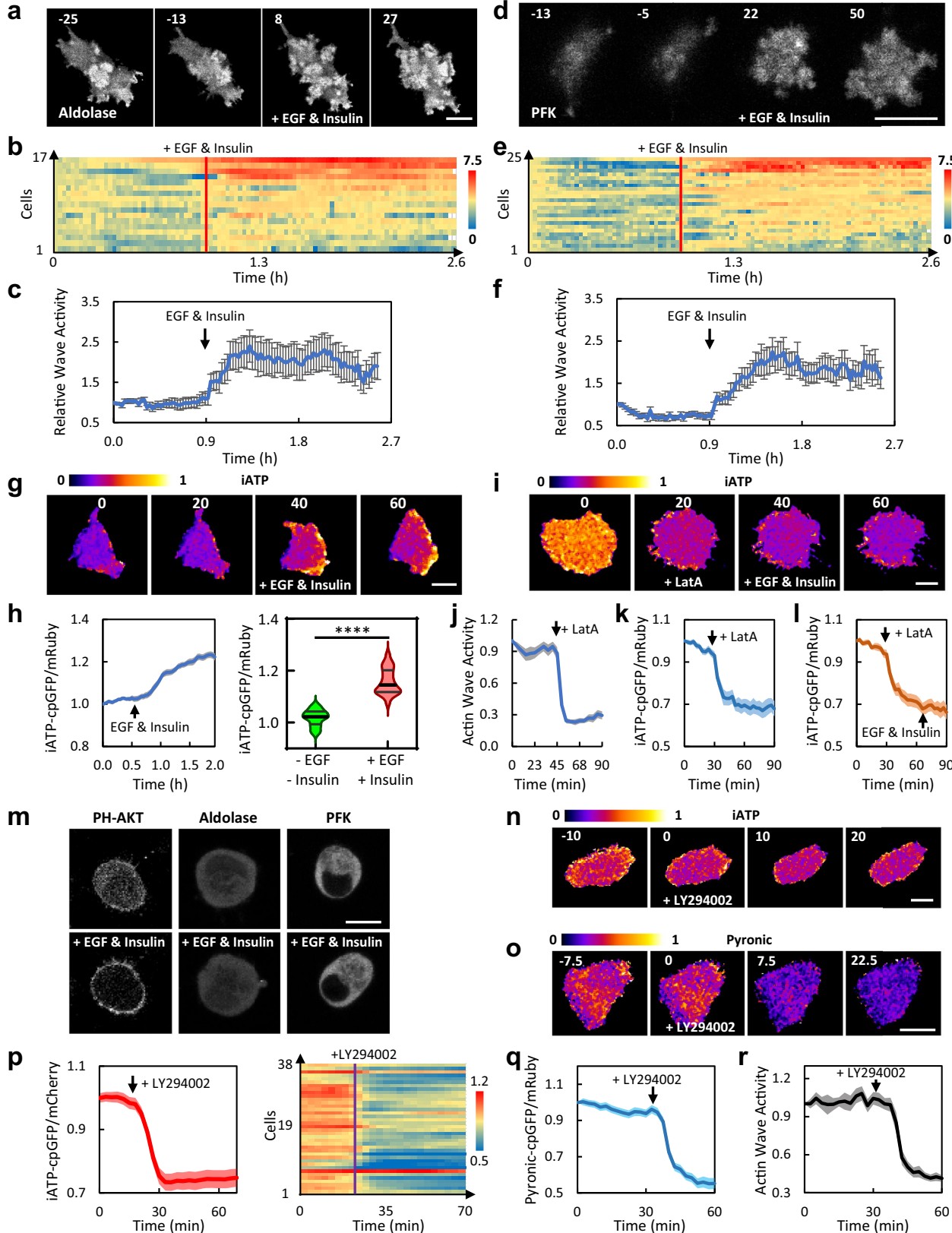

change. Conversely, addition of OXPHOS inhibitors first caused little change in wave activity and further application of glycolysis inhibitors drastically reduced it (Supplementary Fig. 9i), which parallels the ATP change (Supplementary Fig. 4e). The effects of inhibitors were reversible. When they were washed out and cells returned to incubator, we observed normal morphology the next day. We also found that its

catalytic activity is not required for the enrichment of aldolase in the waves since a catalytically dead mutant still localized to the waves (Supplementary Fig. 10).

How can recruitment of a single glycolytic enzyme to the plasma membrane enhance overall cell motility and dynamics? To further investigate this surprising result, we examined how membrane

**Fig. 4 | Effects of perturbing glycolytic waves on overall glycolytic activities.**
**a** Time-lapse confocal images of the basal surface of an MCF-10A M3 cell expressing aldolase-GFP stimulated with 20 ng/ml EGF and 10 μg/ml insulin at 0 min (also see Supplementary Movie 9). **b** Wave activity, defined as areas of pixels with above threshold aldolase-GFP intensity, of cells stimulated with EGF and insulin (also see Supplementary Movie 9). The wave activity normalized to that of the first frame was plotted over 2.6 h for 17 individual cells. Cells are ordered based on their average wave intensity. **c** Plot of normalized wave activity (mean ± SEM) over time for the 17 cells in (**b**). **d** Time-lapse confocal images of the basal surface of an MCF-10A M3cell expressing PFK-GFP stimulated with 20 ng/ml EGF and 10 μg/ml insulin at 0 min (also see Supplementary Movie 9). **e** Wave activity, defined as areas of pixels with above threshold PFK-GFP intensity, of cells stimulated with EGF and insulin. The wave activity normalized to that of the first frame was plotted over 2.6 h for 25 individual cells. Cells are ordered based on their average wave intensity. (**f**) Plot of normalized wave activity (mean ± SEM) over time for the 25 cells in (**e**). **g** iATP cpGFP/mRuby ratio images of an MCF-10A M3 cell stimulated with EGF and insulin at 30 min. A graphic explanation of the ATP sensor's mechanism and design is shown in Supplementary Fig. 4a. **h** Left panel: plot of normalized iATP cpGFP/mRuby (mean ± SEM) over time for 21 cells stimulated with EGF and insulin. The dynamic changes in iATP cpGFP/mRuby of these 21 individual cells were plotted in Supplementary Fig. 6a. Right Panel: violin plots (with quartiles and median) illustrating the average iATP cpGFP/mRuby ratios within 0.5 h imaging windows before and after EGF and insulin stimulation in these 21 cells. ****$p < 0.0001$ (Two-tailed paired t test). **i** iATP cpGFP/mRuby ratio images of an MCF-10A M3 cell treated with 10 μM Latrunculin A (LatA) at 10 min followed by stimulation with EGF and insulin at 30 min. **j** Actin wave activity, defined as ratio of membrane to cytosol LifeAct intensity, was plotted (mean ± SEM) over time for 16 cells before and after

treatment with 10 μM LatA. **k** Plot of normalized iATP cpGFP/mRuby (mean ± SEM) over time for 20 cells treated with LatA. The activity of individual cells was plotted in Supplementary Fig. 6b. **l** Plot of normalized iATP cpGFP/mRuby (mean ± SEM) over time for 23 cells treated with LatA followed by EGF and insulin. The activity of individual cells was plotted in Supplementary Fig. 6c. **m** Confocal images of PH-AKT-RFP, aldolase-GFP, and PFK-GFP before and after treatment with EGF and insulin in MCF-10A M3 cells pretreated with 10 μM LatA. **n** iATP cpGFP/mRuby ratio images of an MCF-10A M3 cell treated with 50 μM LY294002 at 0 min. **o** Pyronic cpGFP/mRuby ratio images of an MCF-10A M3 cell treated with 50 μM LY294002 at 0 min. A graphic explanation of the pyruvate sensor's mechanism and design is shown in Supplementary Fig. 6d. Images of other channels of this cell are shown in Supplementary Fig. 6e. Also see Supplementary Movie 11 (top row). For images of change in NADH/NAD+ biosensor upon treatment with LY294002, see Supplementary Fig. 6g, h, and Supplementary Movie 12. **p** Plot of normalized iATP cpGFP/mRuby (mean ± SEM) over time for 38 cells treated with 50 μM LY294002. Dynamic changes of individual cells over time are plotted in the right panel. **q** Plot of normalized pyronic cpGFP/mRuby (mean ± SEM) over time for 23 cells treated with LY294002. Activities of individual cells are shown in Supplementary Fig. 6f. **r** Actin wave activity, defined as ratio of membrane to cytosol LifeAct intensity, was plotted (mean ± SEM) over time for 16 cells treated with 50 μM LY294002. A representative cell is shown in Supplementary Movie 11 (bottom row). Cells showing changes in iATP biosensor upon treatment with EGF and Insulin, LatA, and LY 294002 were all pretreated with OAR (5 μM Oligomycin, 1 μM Antimycin A, and 1 μM Rotenone) to rule out the ATP change from OXPHOS. Scale bar is 20 μm for all except (**m**) (10 μm). All the cells plotted and quantified in each panel were from at least three independent experiments, which is consistent throughout the later figures.

recruitment of PFK affects aldolase localization by co-expressing aldolase-GFP in cells carrying the PFK recruitment system (Fig. 6a). Consistent with the above findings, recruitment of PFK by CID caused cell spreading and an enhanced level of dynamic protrusions at the perimeter, indicative of increased wave generation (Fig. 6b, Supplementary Fig. 11a, and Supplementary Movie 16). Remarkably, aldolase was also recruited to the plasma membrane (Fig. 6c–f, Supplementary Fig. 11, and Supplementary Movie 16). The coordinated behavior of the two enzymes could indicate that PFK recruitment initiated waves and aldolase was recruited to them or, alternatively, the enzymes are in a complex. In either case, the coordinated recruitment of glycolytic enzymes can explain why the recruitment of one glycolytic enzyme is sufficient to enhance dynamic actin patches and protrusive activities.

## Enhanced glycolytic waves may explain energy shift in cancer cells

The cartoon in Fig. 6g summarizes our conclusions so far indicating that the association of glycolytic enzymes with traveling waves enhances local glycolysis activities, which in turn fuel the formation of further waves. We showed previously that oncogenic transformation with Ras leads to an increased number of dynamically active actin waves[14,15]. Cancer cells depend more on glycolysis as their energy source even when oxygen is available, a phenomenon known as the Warburg effect. Our current study shows that the enrichment of the glycolytic enzymes in the actin waves provides a large fraction of the ATP in cells. Thus, an increased number of waves, and associated glycolytic enzymes, in oncogenically transformed cells might underlie the Warburg effect.

To test the idea, we took advantage of a series of MCF10A derived cell lines, M1-M4, marked by increased oncogenic and metastatic potential[23–25]. While M1 and M2 represent wild-type and Ras-transformed cells, respectively, M3 and M4 cells were further selected for their higher malignancy and metastatic index. We found previously the increasing metastatic potential in this series of cells correlates closely with a sequential increase in wave activities (Fig. 7a). We also expressed the iATP biosensor in M1-M4 cells. As depicted in Fig. 7a, the M1-M4 series of cells with increasing metastatic potential showed a simultaneously increasing level of glycolytic ATP as reflected by cpGFP/mRuby ratio in the OAR pretreated cells. Given that

inhibition of PI3K significantly abrogates wave activity (Fig. 4r), we next treated M1-M4 cells with the PI3K inhibitor. Inhibiting the waves led to a ~ 25% decrease in the cpGFP/mRuby ratio of iATP signal in M3 cells but only ~15% decrease in M1 cells (Fig. 7b, c). Therefore, inhibiting the waves leads to a more significant reduction in glycolytic ATP production in cancer cells compared to non-cancerous parental cells. Thus, the progressively augmented wave activity, and implied greater actin wave associated glycolytic enzymes, seen in the M1-M4 series may provide a mechanism for the greater reliance of the more metastatic cells on ATP from glycolysis.

To further test the generality of this principle, we examined additional cancer cell lines, including those from pancreatic (AsPC-1), lung (Calu-6), breast (MCF-7 and MDA-MB-231), colon (HCT116), and liver (SNU-387 and HepG2) cancers. While MCF-7 cells exhibited minimal wave activity with a level between M1 and M2 cells, all other lines showed significant wave activity: HCT116, MDA-MB-231, and AsPC-1 displayed the strongest activity at levels similar to M3 and M4 cells, with SNU-387 and HepG2 showing intermediate activities between M2 and M3 levels, and Calu-6 at level similar to M2 (Fig. 7d, Supplementary Fig. 12, and Supplementary Movie 17). We then measured ATP levels in each cell line pretreated with OAR and found a strong correlation between wave activities and average ATP levels, indicating that higher wave activity leads to increased ATP production from glycolysis across these cancer cell lines (Fig. 7d). We quantified the proportions of ATP derived from glycolysis and OXPHOS in each cell line, noting a serial increase in the fraction of glycolytic ATP and a decrease in the fraction of OXPHOS ATP across these seven cancer cell lines as wave activities increased from low to high (Fig. 7e). Plots of total intracellular ATP with OAR pretreatment (Fig. 7f), glycolytic ATP fraction (Fig. 7g), and OXPHOS ATP fraction (Fig. 7h) relative to wave activities in these cancer cell lines further show that higher wave activities strongly correlate with an increased reliance on glycolysis for ATP production in cancer cells, a key characteristic of the Warburg effect.

## Perturbations that abolish glycolytic waves inhibit processes required for cancer progression

Cancer is characterized by high energy dependent processes such as nutrient uptake and protein synthesis. Therefore, we assessed the

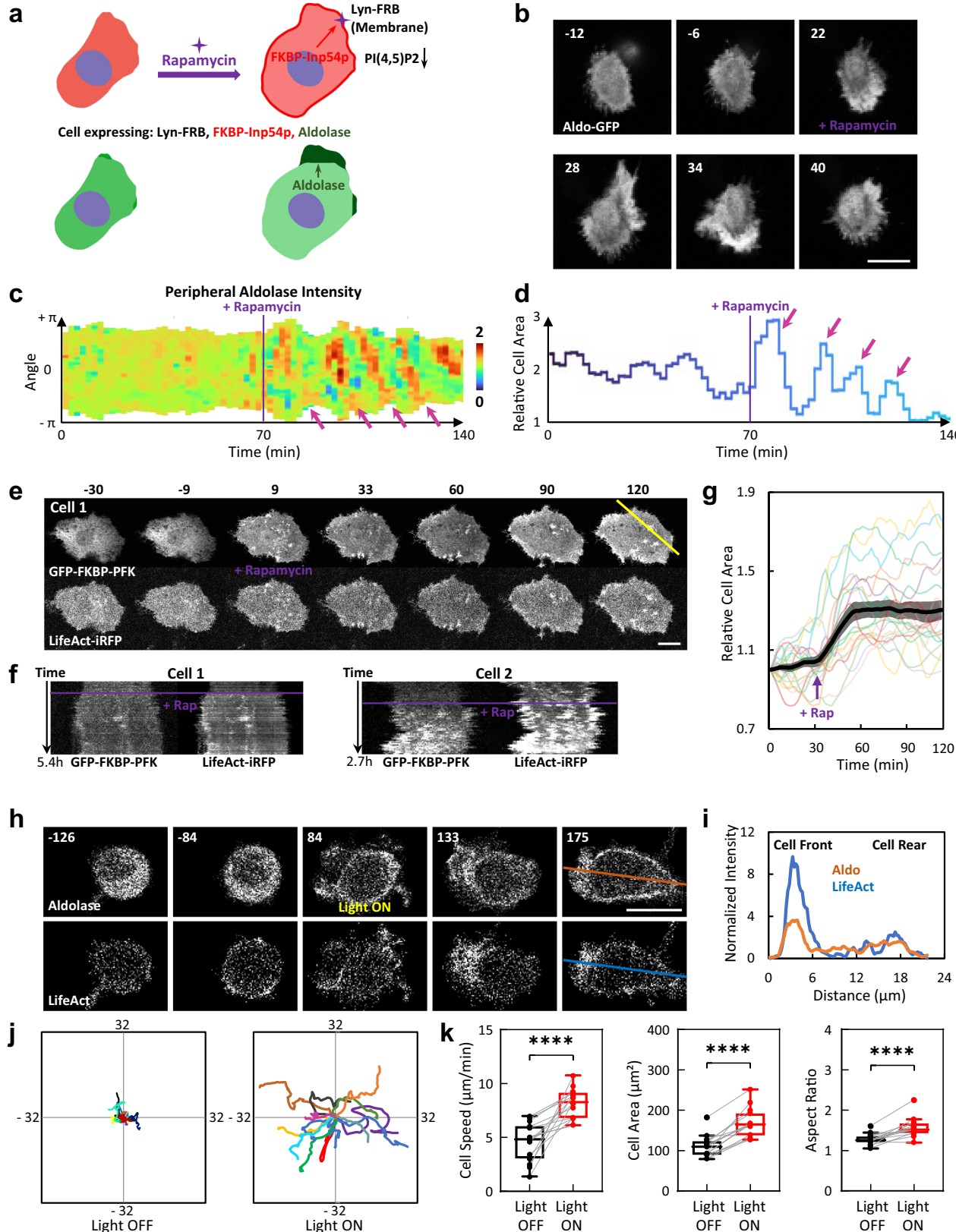

impact of inhibiting ATP production from glycolytic waves on these processes. As shown in Fig. 8a, waves are reduced by approximately 60% with a low dose of DB or LY294002, more severely with a high dose of DB, and are minimally affected by OAR. Likewise, the iATP cpGFP/mRuby levels dropped by about 25% with a low dose of DB or LY, and by more than 70% with a high dose, while OAR had a minor

impact (Fig. 8a). The changes in cellular ATP levels following the inhibition of OXPHOS, glycolysis, and wave activities, as well as stimulation with EGF and insulin, were verified using luciferase-based biochemical measurements (Supplementary Fig. 13). Therefore, we designed a regime of inhibitors to investigate the relationship between ATP from glycolytic waves and physiological processes.

**Fig. 5 | Effects of synthetic and optogenetic perturbations of glycolytic waves on cell dynamic and motility. a** Schematic explanation of the design of experiment in (**b**): addition of rapamycin recruits the FKBP tagged phosphatase Inp54p from cytosol to Lyn-FRB located in the plasma membrane by forming FKBP-rapamycin-FRB complex. Membrane recruited Inp54p then hydrolyzes PI(4,5)P2 and triggers cell spreading and spiral wave formation. The location of aldolase is visualized during this perturbation. More details on the design of the Chemically Inducible Dimerization (CID) system are shown in Supplementary Fig. 7a. Images were created in PowerPoint using licensed elements from BioRender (Created in BioRender. Zhan, H. (2025) https://BioRender.com/g6b620l). **b** Time-lapse confocal images of the aldolase-GFP in an MCF-10A M1 cell expressing CFP-Lyn-FRB, mCherry-FKBP-Inp54p, and aldolase-GFP treated with 1 μM rapamycin at 0 min (also see Supplementary Movie 13). More examples of cells are shown in Supplementary Fig. 7b. **c** Kymograph of aldolase-GFP signal around the perimeter of the cell in (**b**) over time. The pink arrows indicate enrichment of aldolase in the rhythmically spiral waves around the cell perimeter. **d** Quantification of the normalized area of the cell in (**b**) over time. The pink arrows indicate that the cell area change aligns with enrichment of aldolase in the rhythmically spiral waves around the cell perimeter. **e** Time-lapse confocal images of an MCF-10A M3 cell expressing Lyn-FRB, GFP-FKBP-PFK, and LifeAct-iRFP treated with 1 μM rapamycin at 0 min (also see Supplementary Movie 14). **f** Kymographs of the yellow line in (**e**) scanning through FKBP-PFK and LifeAct channels in the cell in (**e**) before and after treatment

with 1 μM rapamycin over 5.4 h. An additional example of another cell is shown on the right panel. **g** Quantification of normalized cell area (mean ± SEM) of n = 20 cells in (**e**) over time. Tracks of changes in individual cells are shown in the color lines. **h** Time-lapse confocal images of a differentiated HL-60 neutrophil expressing CIBN-CAAX, CRY2PHR-mCherry-aldolase and LifeAct-miRFP703, before and after 488 nm light illumination (also see Supplementary Movie 15). Time in sec; scale bar: 10 μm. 488 nm Light is turned on at 0 sec. Schematic explanation of this experiment is shown in Supplementary Fig. 8a. **i** Intensity of aldolase (orange line) and LifeAct (blue line) across the front and rear regions of the cell in (**h**). **j** Centroid tracks of differentiated HL-60 cells showing random motility before and after global recruitment of aldolase. Each track lasts 3 min and was reset to the same origin. n = 15 cells from at least 3 independent experiments. **k** Box-and-whisker plots of HL-60 average cell speed, cell area, and aspect ratio, before (black) and after (red) aldolase recruitment. n = 15 cells from at least 3 independent experiments. ****$p \le 0.0001$ (Two-tailed paired t test). The boxes extend from 25th to 75th percentiles, median is at the center, and whiskers and outliers are graphed according to Tukey's convention. Connecting lines are provided between paired data points obtained from the same cell, before or after aldolase recruitment. Quantifications of non-recruitment control are shown in Supplementary Fig. 8b, c. Scale bar is 10 μm for (**b**) and (**h**), and 20 μm for (**e**). Cell in (**b**) is MCF-10A M1 cell, in (**e**) is MCF-10A M3 cell, and in (**h**) is neutrophil-like HL-60 cell.

---

It has been reported that macropinocytosis is upregulated in cancer transformation[26]. We found macropinocytosis indeed was increased along the M1 – M4 series of cells which are known to have increased metastatic potential. Pre-treatment with a high dose of DB eliminated macropinocytosis in all cell lines, as evidenced by the accumulated uptake of BSA shown in Fig. 8b and Dextran in Supplementary Fig. 14. Treatment with lower concentrations of DB or LY had intermediate effects on all cells (Fig. 8b, c), but the inhibition percentages were significantly higher in M3 and M4 cells compared to M1 and M2 (Fig. 8d). In contrast, OAR did not alter macropinocytosis in any of the cells (Fig. 8b–d and Supplementary Fig. 14).

Subsequently, we measured the rate of protein synthesis using a pulse-chase scheme based on the convertible fluorescent protein: KiK-GR[27]. We noted that the protein synthesis rate was higher in M3 and M4 cells than in M1 and M2, correlating well with increased wave activity and cell proliferation. Using M3 cells to assess the effects of inhibitors, upon 405 nm light stimulation, the green fluorescence of KiK-GR fully converted to red. Over the following 15 h, newly synthesized KiK-GR, marked by green fluorescence, appeared as shown in Fig. 8e, f. The red signal remained relatively stable during this period. A high dose of DB completely halted protein synthesis, while mildly affecting protein stability. Treatment with lower concentrations of DB or LY had a moderate impact on the rate of protein synthesis (Fig. 8e, f). In contrast, OAR had a much weaker effect on protein synthesis when compared to DB or LY (Fig. 8e, f).

To determine if a less energy-dependent biological process is influenced by glycolytic wave activity, we evaluated mitochondrial potential under the same inhibitor regimen[28]. We found that mitochondrial potentials were not significantly different in M1 – M4 cells, as illustrated in Supplementary Fig. 15. Furthermore, high doses of glycolytic inhibitors did not alter mitochondrial potential, whereas OXPHOS inhibitors completely eliminated it.

## Discussion

Our study was prompted by our unexpected discovery that aldolase is associated with propagating waves of Ras/PI3K and F-actin in the cell membrane/cortex. Following this up, we found that all the glycolytic enzymes we tested are associated with the cell waves, co-localized and traveling with the signal transduction and cytoskeletal components. Growth factors and other stimuli increase wave activity and promote the recruitment of glycolytic enzymes into the waves, while inhibitors that abolish waves redistribute the enzymes back to the cytosol.

Consistently, ATP levels are strongly correlated with the augmentation or abrogation of the waves under these various perturbations. Recruitment of a single glycolytic enzyme induces epithelial cell spreading and enhances neutrophils polarity and migration, while inhibition of glycolysis silences the dynamic membrane undulations. Furthermore, synthetic recruitment of PFK causes a co-recruitment of aldolase to the membrane. Crucially, we observed a sequential increase in the frequency of waves and a simultaneous rise in ATP levels across a series of cells exhibiting increasingly higher metastatic indices. Experiments in seven additional cancer cell lines show that cells with higher glycolytic wave activities rely more on glycolysis for ATP production. Energy-intensive processes like macropinocytosis and protein synthesis, which are more active in cancer cells, are strongly impaired by the inhibition of wave activity. This suggests that the heightened presence of glycolytic waves in cancer cells could account for their greater dependence on glycolysis for energy, offering a distinct mechanistic explanation for the Warburg effect.

Although glycolysis is traditionally thought to occur in the cytosol, our findings introduce an unexpected dimension to the understanding of this process. The association of glycolytic enzymes with waves in the cellular cortex/plasma membrane enhances localized ATP production and enables a level of regulation beyond allostery and post-translational modifications. Some previous studies have shown individual glycolytic enzymes enriched in specific subcellular structures, such as the plasma membrane, stress fibers, and condensates, which are relatively static and lack dynamics[29–37]. The propagating waves of glycolytic enzymes described in our study represent a distinct and novel subcellular structure, providing a high level of spatiotemporal regulation of glycolysis and meeting the demand for fast ATP turnover during energy-intensive cellular events such as migration, macropinocytosis, and protein synthesis. Since wave activity tracks with the metastatic potential, it may also be a mechanism of matching metabolism to cellular state.

Our observations raise several questions about the molecular basis and consequences of glycolytic waves. How do the glycolytic enzymes form the self-organized waves? Several glycolytic enzymes have been shown to bind F-actin in vitro[38–43]. The molecular determinant of binding has not been clearly elucidated for the majority of these enzymes, although an actin binding region was identified in aldolase[44]. However, the width of the glycolytic bands is wider than that of the LifeAct, it is possible that the localization of glycolytic enzymes may not be through direct binding to newly polymerized F-actin. Instead, these enzymes

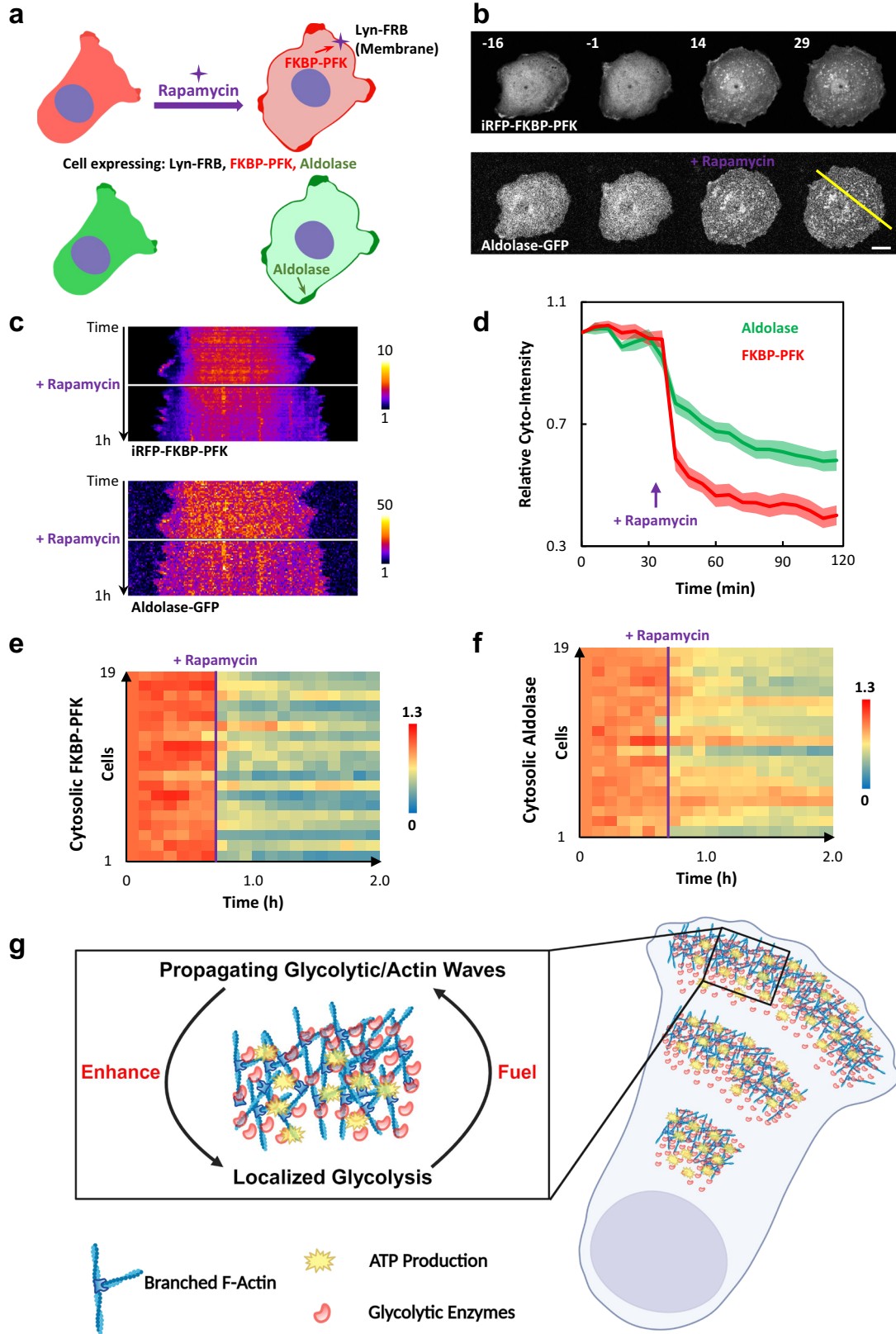

may be associated with other molecules such as signaling molecules, which display similar diffusive wave bands. Importantly, synthetic recruitment of PFK caused a co-recruitment of aldolase. This may suggest a direct association of the enzymes with each other or association with a common scaffold. Elucidating the mechanism regulating the localization of the glycolytic enzymes to the propagating waves may provide new therapeutic targets.

What is the effect of the association of glycolytic enzymes with the waves on the rate of glycolysis? Concentrating the enzymes on the membrane/cortex in these waves may enhance the reaction rate. Based on our imaging studies at least 15% of the enzymes are recruited into the waves. We did not have the resolution to determine whether the enzymes were associated with the membrane or with the cortex. Nevertheless, if a cytosolic protein were concentrated into a 0.5 μm

**Fig. 6 | Synthetic recruitment of PFK to the cell membrane triggers co-recruitment of aldolase. a** Schematic explanation of the design of experiment in (**b**): addition of rapamycin recruits the FKBP tagged PFK from cytosol to Lyn-FRB located in the plasma membrane by forming FKBP-rapamycin-FRB complex. The location of aldolase is visualized and measured during this perturbation. Images were created in PowerPoint using licensed elements from BioRender (Created in BioRender. Zhan, H. (2025) https://BioRender.com/i18w4bj). **b** Time-lapse confocal images of the iRFP-FKBP-PFK and aldolase-GFP channels in an MCF-10A M3 cell expressing Lyn-FRB, iRFP-FKBP-PFK, and aldolase-GFP treated with 1 µM rapamycin at 0 min. Scale bar is 20 µm. An additional example of another cell is shown in Supplementary Fig. 11a. Also see Supplementary Movie 16. **c** Kymographs of iRFP-

FKBP-PFK and aldolase-GFP of the cell in (**b**) across the yellow line over time. **d–f** The means ± SEMs of the normalized intensity for cytosolic iRFP-FKBP-PFK and cytosolic aldolase-GFP from n = 19 cells expressing Lyn-FRB, iRFP-FKBP-PFK, and aldolase-GFP, before and after rapamycin treatment over time, are plotted in (**d**). Data from these 19 individual cells are shown in the color heat maps in (**e**) for FKBP-PFK and in (**f**) for aldolase. **g** Model: enrichment of glycolytic enzymes in self-organized glycolytic/F-actin waves enhance local glycolysis to provide energy for new wave formation, cell migration and other cellular processes. Images were created in BioRender using modified, licensed elements from its library (Created in BioRender. Zhan, H. (2025) https://BioRender.com/4apvlic).

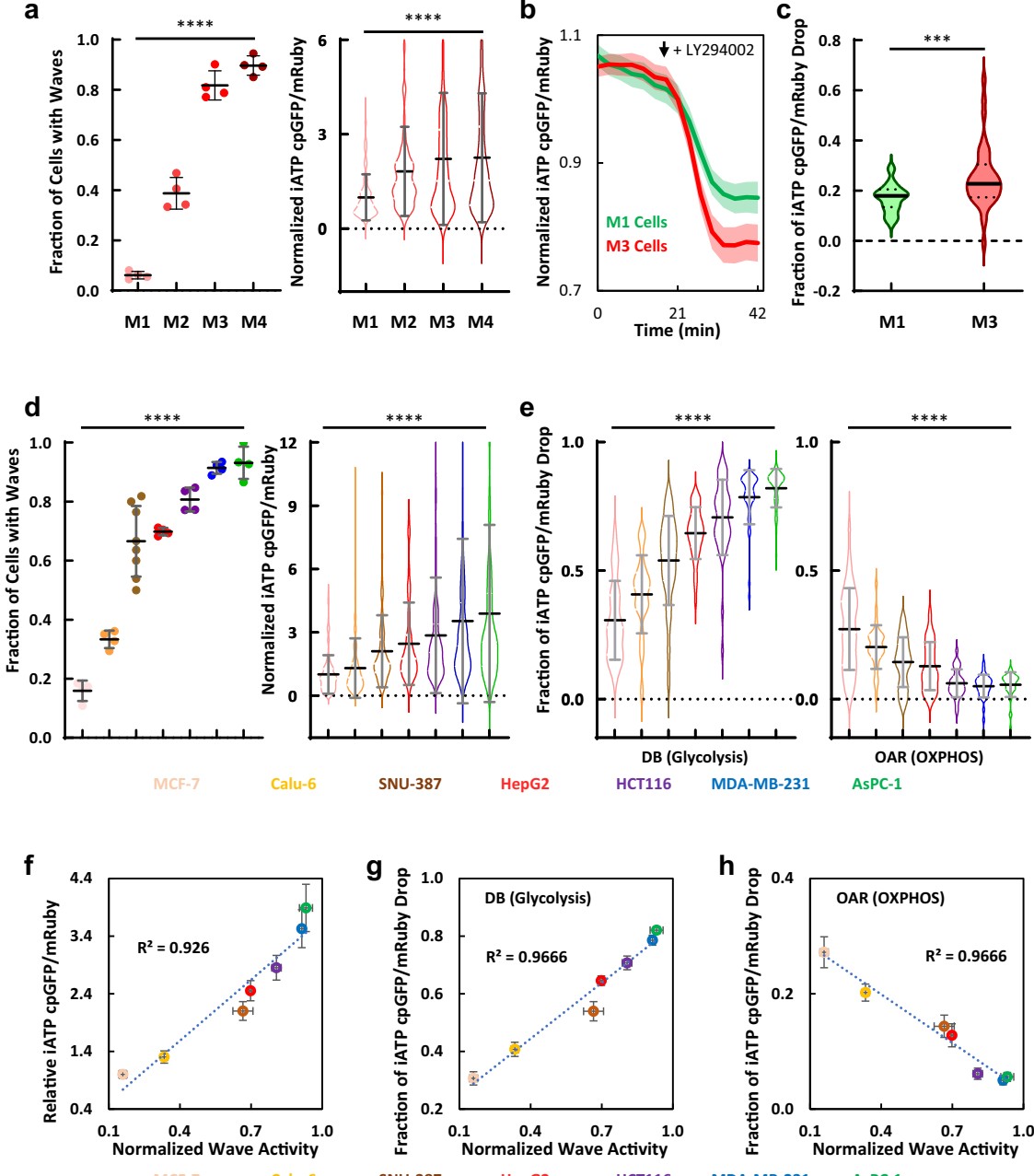

shell covering the entire membrane/cortex of a 30 µm cell, the concentration increase would be 20-fold. Furthermore, the enzymes are recruited into a small area of the cortex covered by waves, so the overall concentration effect is quite significant. If the enzymes associate with the membrane rather than the cortex, the concentration effect could be much higher.

What is the functional significance of glycolytic waves? Our findings suggest that ATP production from these glycolytic waves may constitute approximately 33% of the total ATP generated through glycolysis, given that iATP cpGFP/mRuby levels were reduced by ~25% with LY or LatA treatment and ~75% with high DB. The finding that EGF and insulin stimulation fails to enhance glycolytic ATP in the presence

**Fig. 7 | Glycolytic waves regulate the energy shift in cancer. a** Left: Quantification of wave activity from previous published work[14] is re-plotted for comparison (mean ± SD of fraction of cells with waves during a 2 h imaging window, 501 M1 cells, 608 M2 cells, 234 M3 cells, and 302 M4 cells from 4 independent experiments). Welch's ANOVA test was performed for the four groups, ****$p$ < 0.0001. Right: Quantification of intracellular iATP cpGFP/mRuby ratio, normalized to the mean of M1 cell, in M1-M4 MCF-10A cells pretreated in OAR (violin plot and mean ± SD of 112 M1 cells, 139 M2 cells, 200 M3 cells, and 198 M4 cells from 5 independent experiments). Welch's ANOVA test was performed for the four groups, ****$p$ < 0.0001. **b** Normalized iATP cpGFP/mRuby ratio signal shown in mean ± SEM of $n$ = 27 M1 cells (green) and $n$ = 38 M3 cells (red) from 4 independent experiments treated with LY294002 at the indicated time. The cpGFP/mRuby values are normalized to the time point that LY294002 was added. Cells were pretreated in OAR before the treatment with LY294002. **c** Fraction of iATP cpGFP/mRuby drop upon treatment with LY294002 in these 27 M1 cells (green) and 38 M3 cells (red) in (**b**) were shown as violin plots (with quartiles and median) and compared by two-tailed unpaired t test with Welch's correction, ***$p$ = 0.0007. **d** Left: Quantification of wave activities in 7 cancer cell lines (mean ± SD of fraction of cells with waves during a 2 h imaging window; 289 MCF-7 cells, 208 Calu-6 cells, 108 SNU-387 cells, 221 HepG2 cells, 301 HCT116 cells, 275 MDA-MB-231 cells, and 220 AsPc-1 cells from 4 independent experiments except SNU-387 cells from 8 independent experiments are quantified). Welch's ANOVA test was performed for the 7 groups, ****$p$ < 0.0001. Representative confocal images showing the wave activities in these 7 cell lines are displayed in Supplementary Fig. 12 and Supplementary

Movie 17. Right: Quantification of iATP cpGFP/mRuby ratio, normalized to the mean of MCF-7 cell, in 7 cancer cell lines pretreated in OAR (violin plot and mean ± SD of 175 MCF-7 cells, 177 Calu-6 cells, 110 SNU-387 cells, 124 HepG2 cells, 163 HCT116 cells, 140 MDA-MB-231 cells, and 105 AsPc-1 cells from at least 4 independent experiments). Welch's ANOVA test was performed for the 7 groups, ****$p$ < 0.0001. In both plots, from left to right, the cell lines are: MCF-7, Calu-6, SNU-387, HepG2, HCT116, MDA-MB-231, and AsPc-1, each represented by a unique color. **e** Quantification of fraction of iATP cpGFP/mRuby drop upon treatment with DB (left panel) or OAR (right panel) in 7 cancer cell lines (violin plot and mean ± SD of 45 DB-treated and 35 OAR-treated MCF-7 cells, 38 DB and 33 OAR Calu-6 cells, 27 DB and 25 OAR SNU-387 cells, 37 DB and 22 OAR HepG2 cells, 38 DB and 31 OAR HCT116 cells, 38 DB and 31 OAR MDA-MB-231 cells, and 37 DB and 38 OAR AsPc-1 cells from at least 4 independent experiments). Welch's ANOVA was performed for the seven cell lines upon the treatment with DB (left) or OAR (right), respectively; ****$p$ < 0.0001. In both plots, from left to right, the cell lines are: MCF-7, Calu-6, SNU-387, HepG2, HCT116, MDA-MB-231, and AsPc-1, each represented by a unique color. **f–h** The mean ± SEM of intracellular ATP levels in cells pretreated with OAR as shown in (**d**), and the mean ± SEM of ATP reduction fractions in cells upon treatment of DB or OAR as shown in (**e**), are plotted against the mean ± SEM of wave activities among the 7 cancer cell lines from (**d**), resulting in the new graphs (**f**), (**g,h**), respectively. Linear regression trend lines and corresponding $R^2$ values are shown in each plot. Cell lines from left to right in all three plots are MCF-7, Calu-6, SNU-387, HepG2, HCT116, MDA-MB-231, and AsPc-1, each indicated by a unique color.

of LatA (Fig. 4l) suggests that the boost in glycolytic ATP by growth factors specifically acts through glycolysis in these waves, rather than through cytosolic glycolysis. The synthetic recruitment of glycolytic enzymes to the plasma membrane significantly enhances morphological dynamics and cell movement (Figs. 5, 6), offering further evidence that these effects are specifically fueled by the increase in glycolytic waves rather than by an enhancement of overall cytosolic glycolysis, as the abundance of glycolytic enzymes in the cytosol is reduced upon synthetic recruitment. Our study also suggests that local ATP production from these glycolytic waves in the cell periphery may provide the energy that fuels actin wave formation. Existing studies have shown that *Dictyostelium*, oocytes, mast cells, epithelial cells, neurons, and other cells display these actin waves, which regulate many biological processes such as cell growth, cell cycle, phagocytosis, protein trafficking, synaptogenesis, and migration in these cells[45–62]. Glycolytic waves may control these functions as well.

Our study provides a potential explanation for the Warburg effect. Tumor cells often reprogram their energy metabolism by using glycolysis as the main source of ATP production rather than oxidative phosphorylation, even when oxygen is available[3,63–65]. This effect, known as aerobic glycolysis or the Warburg effect, was reported by Otto Warburg nearly 100 years ago[4,5], yet the mechanism is not fully understood. We have previously suggested that cancer cells are shifted to a lower threshold "state" of key signal transduction and cytoskeletal networks, such as the Ras/PI3K/ERK network involved in oncogenic transformation[14]. Our current study shows that the increased association of the glycolytic enzymes with the cortex/membrane accompanies a shift in the "state" of the excitable networks. Furthermore, by lowering the threshold via decreasing levels of PI(4,5)P2, we were able to trigger additional recruitment of aldolase, while simultaneously activating changes in cell morphology. We demonstrated that oncogenic transformation leads to increased glycolytic waves, resulting in higher order spatiotemporal organization of glycolytic activity to fuel energy-intensive processes such as motility, macropinocytosis and protein synthesis which are likely to exacerbate cancer progression and metastasis. The property of glycolytic enzymes enriched as self-organizing dynamic waves provides an elegant mechanism for enhancing glycolytic activity to meet the high demand for rapid ATP supply in the energy-intensive dynamic cellular activities associated with cancer progression.

## Methods

### Cells
M1 (MCF-10A), M2 (MCF-10AT1k.cl2), M3 (MCF-10CA1h), and M4 (MCF-10CA1a.cl1) cells, purchased from the Animal Model and Therapeutic Evaluation Core (AMTEC) of Karmanos Cancer Institute of Wayne State University, were all grown at 37 °C in 5% $CO_2$ using DMEM/F-12 medium (Gibco, #10565042) supplemented with 5% horse serum (Gibco, #26050088), 20 ng/ml EGF (Sigma, #E9644), 100 ng/ml cholera toxin (Sigma, #C-8052), 0.5 mg/ml hydrocortisone (Sigma, #H-0888) and 10 μg/ml insulin (Sigma #I-1882). Human neutrophil-like HL-60 cells were gifted by Orion Weiner (UCSF) and grown in supplemented RPMI medium 1640 (Gibco #22400-089) as described previously[66,67]. For cell differentiation, neutrophils were incubated with 1.3% DMSO for 5–7 days before experimentation[67,68].

MCF-7 cells were maintained in EMEM media (Quality Biological, #112-018-101) supplemented with 10% FBS (Gibco, #16140071), 1% penicillin and streptomycin (Pen/Strep) (Sigma, #P0781-100ML), and 10 μg/ml insulin. Calu-6 and HepG2 cells, gifts from Jun Liu lab (JHU), were maintained in EMEM media supplemented with 10% FBS and 1% Pen/Strep. HCT116 cells, gifts from Jun Liu lab (JHU), were maintained in McCoy's 5a Medium (Thermo Fisher, #16600082) supplemented with 10% FBS and 1% Pen/Strep. MDA-MB-231 and SNU387cells, gifts from Jun Liu lab (JHU), were maintained in DMEM media supplemented with 10% FBS and 1% Pen/Strep. AsPC-1 cells, gifts from Douglas Robinson lab (JHU), were maintained in RPMI1640 media supplemented with 10% FBS, 1% Pen/Strep, 1 mM sodium pyruvate (Gibco, #11360070), 1 x nonessential amino acids (Gibco, #11140050), and 10 μg/ml insulin. Cell lines were grown for no more than 10 passages in all experiments.

### Plasmids
Constructs of CFP-Lyn-FRB, and mCherry-FKBP-Inp54p were obtained from Inoue Lab (JHU). GFP/RFP-PH-AKT and RFP-LifeAct were obtained from Desiderio Lab (JHU). Aldolase-GFP was generously provided by the Wulf Lab (Harvard)[31]. PFK-GFP (#116940)[33], Truncated-HK-GFP (#21918)[69], Lifeact-iRFP (#103032)[70], mRuby3-iATPSnFR1.0 (#102551)[19], PyronicSF-mRuby (#124830)[21], Peredox-mCherry (#32380)[22], and Kik-GR (#32608)[27] constructs were obtained from AddGene. Enolase-RFP, PK-RFP, GAPDH-RFP, GFP/iRFP-FKBP-PFK, and Aldolase (D34S)-GFP were generated in this study.

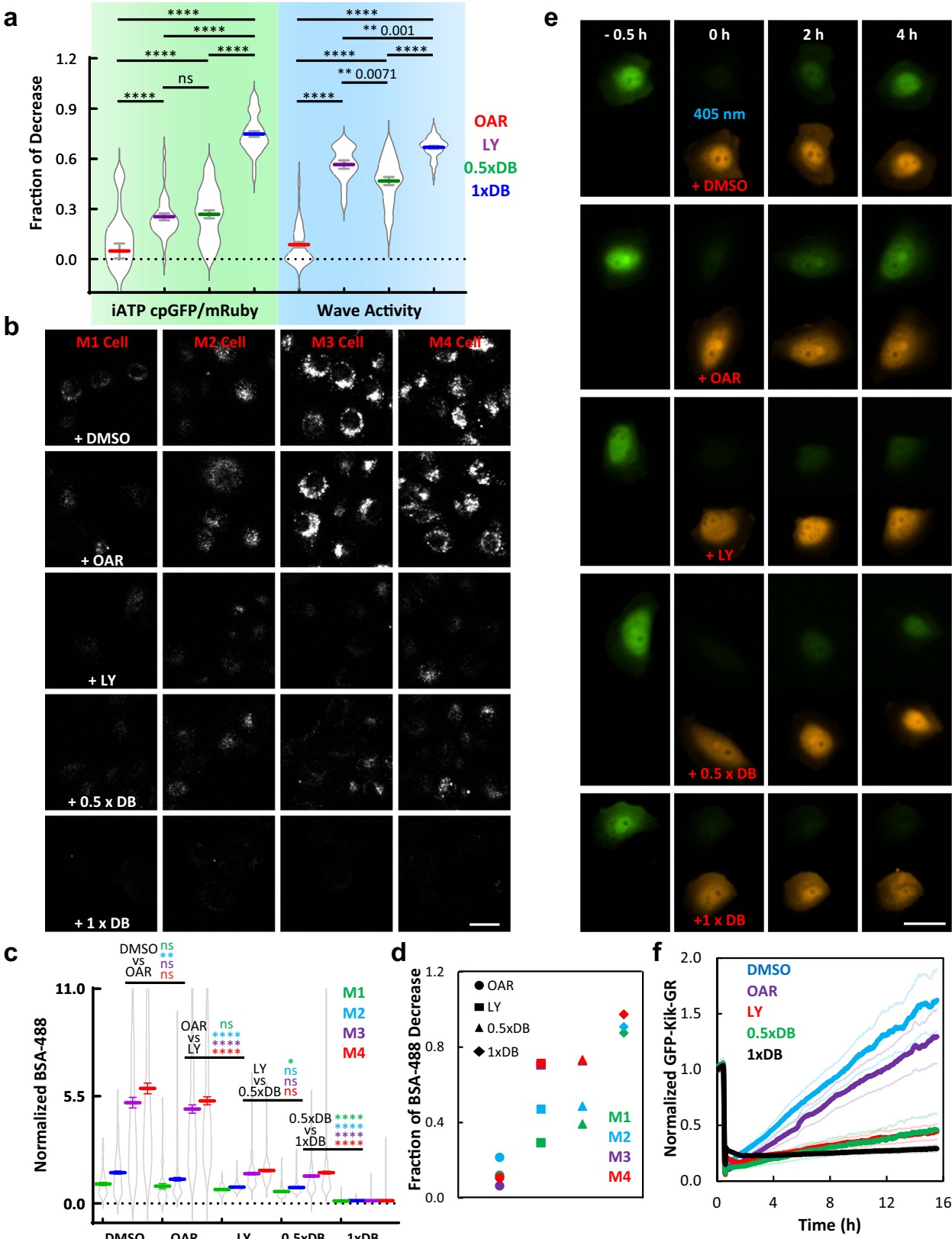

Lyn-FRB and FKBP-Inp54p were subcloned into the lenti-viral expression plasmid pFUW2. 3^rd generation lentiviral constructs, CIBN-CAAX/pLJM1 and LifeAct-miRFP703/pLJM1, were generated in a previous study[66]. The human aldolase A ORF (1092 bases) was PCR-amplified and cloned into BspEI/SalI sites of the PiggyBac™ transposon plasmid to generate the CRY2PHR-mCherry-Aldolase A/pPB construct.

All constructs were verified by sequencing at the JHMI Synthesis and Sequencing Facility.

**Drugs**

Stocks of 25 mM Latrunculin A (Enzo, #BML-T119-0100), 50 mM LY294002 (Invitrogen, #PHZ1144), 10 mM Oligomycin (Cell Signaling

**Fig. 8 | ATP generated by glycolytic waves controls physiological processes associated with cancer progression. a** Quantification of the fraction of iATP cpGFP/mRuby drop (green shade) in MCF-10A-M3 cells upon treatment with OAR (red), LY (purple), 0.5 x DB (green), or 1 x DB (blue) (mean ± SEM of 46, 38, 45, and 47 cells of each treatment condition respectively from at least 4 independent experiments). Quantification of the fraction of actin wave activity decrease (blue shade) in MCF-10A-M3 cells upon treatment with OAR (red), LY (purple), 0.5 x DB (green), or 1 x DB (blue) (mean ± SEM of fraction drop in the ratio value of membrane to cytosol LifeAct intensity of 56, 16, 33, and 56 cells of each treatment condition respectively from at least 4 independent experiments). Unpaired t test with Welch's correction, ****$p < 0.0001$, **$p < 0.01$, ns not significant. Drug concentrations are the same as indicated in the previous experiment, while 0.5 x DB is 5 mM 2-Deoxy-D-glucose and 25 uM 3-Bromopyruvic acid, half of the concentration of 1 x DB used in the previous assays. **b** Confocal images showing the uptake of fluorescence conjugated BSA by the M1-M4 cells treated with DMSO, OAR, LY, 0.5 x DB, or 1 x DB, respectively. The scale bar is 20 μm. The confocal images with the similar experimental design representing the uptake of fluorescence conjugated dextran by cells is shown in Supplementary Fig. 14. **c** Quantification of average cellular BSA uptake value (total intracellular BSA fluorescence intensity divided by the cell size, all normalized to the population average of M1 cells in DMSO control group) of M1-M4 cells in the treatments with DMSO, OAR, LY, 0.5 x DB, or 1 x DB, respectively (violin plot and mean ± SEM of 63 M1, 178 M2, 263 M3, and 304 M4 cells treated with DMSO, 50 M1, 196 M2, 263 M3, and 301 M4 cells treated with OAR, 46 M1, 98 M2, 126 M3, and 155 M4 cells treated with LY, 85 M1, 188 M2, 280 M3, and 268 M4 cells treated with 0.5 x DB, 84 M1, 188 M2, 280 M3, and 268 M4 cells treated with 1 x DB from at least 4 independent experiments). Unpaired t test with Welch's correction, ****$p < 0.0001$, **$p = 0.0044$, *$p = 0.0468$, ns not significant. The statistical differences of BSA uptake between different treatment conditions are shown in green, blue, purple, and red statistical symbols ("*" or "ns"), for M1-M4 cell lines, respectively. **d** The fraction of the difference of cellular BSA uptake mean value shown in (c) between each inhibitor treatment and DMSO control in M1-M4 cell lines are plotted in green, blue, purple, and red, respectively. **e** Epifluorescence images of green (upper) and red (lower) channels showed the fluorescence intensity of Kik-GR in M3 cells over time before and after treatment with DMSO, OAR, LY, 0.5 x DB, or 1 x DB, respectively. 405 nm light was turned on for 5 s of two cycles at time 0 for photo-conversion; the scale bar is 20 μm. Drug treatments were applied at time 0 right after the photo-conversion. **f** Quantification of the fluorescence intensity of the green channel of Kik-GR in M3 cells over 16 h before and after treatment with DMSO (blue line), OAR (purple line), LY (red line), 0.5 x DB (green line), or 1 x DB (black line), respectively (mean ± SEM of 35 cells in DMSO, 37 cells in OAR, 51 cells in LY, 45 cells in 0.5 x DB, and 50 cells in 1 x DB from at least 4 independent experiments). Photo-conversion was performed, and drug treatments were applied at time 0.5 h. The thick line represents the mean, while the upper and lower thin lines in the same color represent the associated mean ± SEM.

Technology #9996 L), 5 mM Antimycin A (Sigma, #A8674), 5 uM Rotenone (Sigma, #R8875), 20 mM MitoTracker (Thermo Fisher, #M7514), 5 mg/ml JC-1 (Thermo Fisher, #T3168), 5 mg/ml BSA-488 (Thermo Fisher, #A13100), 100 mg/ml TRITC-Dextran (Sigma, #T1162-100MG) and 10 mM Rapamycin (Cayman, #13346) were prepared by dissolving the chemicals in DMSO. 400 mM 2-Deoxy-D-Glucose (BioVision, #B1048) and 400 mM 3-Bromopyruvic Acid (BioVision, #B1045) were prepared by dissolving the chemicals in sterile ddH2O. The EGF stock solution was prepared by dissolving EGF (Sigma, #E9644) in 10 mM acetic acid to a final concentration of 1 mg/ml. Insulin (Sigma #I-1882) was resuspended at 10 mg/ml in sterile ddH2O containing 1% glacial acetic acid. Hydrocortisone (Sigma #H-0888) was resuspended at 1 mg/ml in 200 proof ethanol. Cholera toxin (Sigma #C-8052) was resuspended at 1 mg/ml in sterile ddH2O and stored at 4 °C. 4% fresh paraformaldehyde (PFA) was prepared in 1xPBS from 16% methanol-free PFA (Thermo Scientific, #28906, stored in room temperature). All drug stocks except cholera toxin and PFA were stored at −20 °C. The stocks were diluted to the indicated final concentrations in culture medium or live cell imaging medium during experiments.

### Antibodies
Primary antibodies against GFP (Invitrogen, #33-2600), GAPDH (Invitrogen, #MA5-15738), Aldolase A (Sigma-Aldrich, #HPA004177), Enolase 1 (Proteintech, #11204-1-AP), PFKP (Sigma-Aldrich, # HPA018257), and HK-1 (Proteintech, #19662-1-AP). Secondary antibodies: Goat anti-Mouse IgG (H + L) Alexa Fluor™ 660 (Invitrogen, #A-21055), Goat anti-Rabbit IgG (H + L) Alexa Fluor™ 633 (Invitrogen, # A-21071), Goat anti-Mouse IgG (H + L) Alexa Fluor™ 488 (Invitrogen, #A-11001), and Goat anti-Rabbit IgG (H + L) Alexa Fluor™ 488 (Invitrogen, #A-11034).

### Growth factors stimulation
For EGF and Insulin stimulation assays, MCF-10A (M1 - M4) and MDA-MB-231 cells were starved in pure DMEM/F-12 and DMEM medium for 24 h before stimulation.

### Transient transfection
Transient transfections of the cells were performed using Lipofectamine 3000 (Invitrogen, #L3000008) following manufacturer's instructions. Cells were maintained in 35 mm glass-bottom dishes (Mattek, #P35G-0.170-14-C) or chambered coverglass (Lab-Tek, #155409PK) and allowed to attach overnight prior to imaging. Cells were seeded and incubated at 37 °C in 5% CO$_2$ overnight before live cell imaging.

### Virus generation
Cell Seeding and Transfection: On day 1, 293 T cells were seeded at a density of 6×10^5 cells/ml in 25 ml of culture medium into 15 cm cell culture dishes. The next day, conventional calcium phosphate transfection was employed to introduce expression and packaging plasmids into the 293 T cells. The transfection mixture consisted of 20 μg of pFUW2, 9.375 μg each of pMDL, pRSV, and pCMV plasmids, combined with 250 μl of CaCl2 and sterile deionized water, bringing the total volume to 2.5 ml. This mixture was then combined with 2.5 ml of 2x HEPES buffer (pH 7.05) and incubated for 5 min. The resulting transfection mixture was gently added to the plated cells. After 4−6 h, the medium was replaced with fresh culture medium.

Virus Collection and Concentration: On day 5, the medium containing viral particles was harvested from the transfected cells. It was initially centrifuged at 3000 g for 3 min to remove cellular debris and then filtered through a 0.45 μm filter. Subsequently, the filtrate was subjected to ultracentrifugation at 82,700 g for 90 min at 4 °C using a Beckman ultracentrifuge with SW 28 Swinging-Bucket Rotor. The supernatant was discarded, and the viral pellet was resuspended in 70 μl of phosphate-buffered saline (PBS) and incubated overnight at 4 °C for virus concentration. The concentrated virus was aliquoted into 25 μl portions and stored at −80 °C.

### Stable line generation
A stable HL-60 cell line co-expressing CIBN-CAAX (untagged) and LifeAct-miRFP703 was generated using a lentiviral-based approach described in previous studies[66]. In this dual expressing cell line, we stably expressed CRY2PHR-mCherry-AldolaseA via transposon-based integration[66,67,71].

### Macropinocytosis measurement
Cells were incubated with each indicated inhibitors (DMSO, OAR, LY, 0.5 x DB or 1 x DB) for 0.5 h, then either 0.2 mg/ml BSA-488 or 4 mg/ml TRITC-Dextran was added to further incubate for 1 h (BSA group) or 2 h (Dextran group) in temperature and CO$_2$ controlled incubator. Cells were immediately washed with 3 times of PBS and set up for live-cell imaging in the conventional culture medium after the BSA or Dextran incubation.

## Protein synthesis measurement

Cells were maintained in incubator for 24 h after the Kik-GR transfection. Live cell imaging was acquired using Nikon Ti2-E microscope with an iLas2 Ring-TIRF module. The angle was set at 0 degrees to create a wide-field fluorescence image. Cells were maintained in the CO2, humidity, and temperature-controlled chamber during the long-term live imaging. Cells were firstly imaged for 3 min of 10 cycles, then 2 times of 5-second exposure to 405 nm light were applied to each acquisition positions, and different inhibitors were added to cells in each imaging positions respectively right after the photo-conversion of Kik-GR. GFP and RFP channels were continuously acquired for another 300 cycles with 3-min interval following photo-conversion and drugs application.

## Mitochondria potential assessment

JC-1 dye, a commonly used indicator for mitochondrial potential[72], was applied to cells for 20 min at the concentration of 5 μg/ml. Cells were maintained in the incubator during incubation with JC-1 before being rinsed with 3 times of PBS and switched back to normal medium for the setting up of live-cell imaging in Zeiss LSM 780 or LSM880 confocal microscopy. Two emissions were collected (Green: 515–545 nm, and Red: 570–610 nm) both with 514 nm excitation. Different inhibitors were applied during live imaging acquisition.

## Microscopy

All live-cell imaging was performed in a microscope incubation chamber maintaining controlled temperature, humidity, and $CO_2$ levels. All time-lapse live-cell imaging experiments were carried out using one of the following microscopes: 1) Zeiss LSM 780-FCS single-point laser-scanning confocal microscope (Zeiss Axio Observer with 780-Quasar; 34-channel spectral, high-sensitivity gallium arsenide phosphide (GaAsP) detectors); 2) Zeiss LSM 880-Airyscan FAST super-resolution single-point confocal microscope (Zeiss Axio Observer with 880-Quasar; 34-channel spectral, high-sensitivity GaAsP detectors); 3) Zeiss LSM 800 GaAsP single-point laser-scanning confocal microscope with a wide-field camera; 4) Nikon Eclipse Ti-E dSTORM total internal reflection fluorescence (TIRF) microscope (Photometrics Evolve EMCCD camera). The Zeiss LSM 780-FCS and LSM 880-Airyscan microscopes were controlled using ZEN Black software, whereas the Zeiss LSM 800 was controlled using ZEN Blue software. The Nikon TIRF microscope was operated using NIS-Elements software. For imaging, the Zeiss confocal microscopes employed 40× or 63× oil-immersion objectives (with appropriate digital zoom), while the Nikon TIRF system used a 100×/1.4 NA Plan-Apochromat oil-immersion objective. For the Zeiss confocal systems, 405 nm or 445 nm excitation was used for CFP and BFP, 488 nm for GFP and YFP, 561 nm for RFP and mCherry, and 633 nm for far-red fluorophores. For the Nikon TIRF microscope, 488 nm excitation was used for GFP, 561 nm for mCherry and RFP, and 640 nm or 647 nm for far-red fluorophores. For imaging protein synthesis, lasers were directed at 90° incidence to generate wide-field fluorescence images in a Nikon Ti2-E microscope equipped with an ILas2 ring-TIRF module.

## Cell fixation and immunofluorescent staining

For cell fixation during live cell imaging (Fig. 3a and Supplementary Fig. 3), 1 volume of 16% PFA was added to 3 volume of imaging medium to the cells during time-lapse acquisition of Zeiss confocal 780 or 880.

For immunostaining, growth medium was removed and the cells were fixed for 15 min in 4% fresh paraformaldehyde (PFA) prepared in PBS from 16% methanol-free PFA. After permeabilization for 10 min with 0.1% Triton X-100 in PBS, the cells were later washed with PBS once and incubated for 1 h at room temperature in blocking buffer (4% FBS in PBS) to block non-specific sites. Cell were incubated overnight at 4 °C with primary antibodies in blocking buffer. The following day the cells are washed 4 × 10 min with PBS and incubated for 2 h at room temperature with secondary antibodies in blocking buffer in dark room. The cells were later counterstained for 5 min with the nuclear stain DAPI at 1 ug/mL in PBS, washed 4 × 10 min with PBS, and mounted in PBS containing 0.02% (w/v) NaN3 in dark room. The cells were later imaged in Zeiss confocal 780 for glycolytic enzymes stained by antibodies (secondary antibodies are conjugated with either Alexa® Fluor 660/633 or 488) and LifeAct-RFP channel.

## Biochemical measurement of ATP

The luciferase based cellular ATP measurement was modified from the suggested protocol of Abcam by using its Luminescent ATP Detection Assay Kit (#ab113849). In a sterile white 96-well plate, cells were seeded a day before and the confluency was about 70% on the day of experiment. ATP standard was prepared at 0 (growth media), 0.001 μM, 0.01 μM, 0.1 μM, 1 μM, 10 μM, 100 μM, and 1000 μM, with two replicates. Cells were treated with DMSO, LY294002, LatA, DB, or OAR, for 40 min before lysis buffer (detergent) was added. For EGF & Insulin stimulation, cells were first starved in pure DMEM/F-12 medium for 24 h, and later lysed by lysis buffer (detergent) after 40 min incubation with EGF & Insulin. After the addition of detergent, the plate of the cells was sealed and shaken for 5 min in an orbital shaker at 300 rpm before substrate solution was added to each of the wells. The plate was sealed and shaken again for 5 min in an orbital shaker at 300 rpm, and adapted in dark for 10 min before imaging in the luminescence plate reader (BMG Labtech Omega). Every treatment condition was performed with at least two replicates.

## Optogenetic experiments

Optical experiments were done without chemoattractant. Photo-activation was performed with a 488 nm excitation Argon laser, CRY2PHR-mCherry-Aldolase A was visualized with a 561 nm excitation solid-state laser, and LifeAct-miRFP703 was excited with a diode laser (633 nm excitation). A 40X/1.30 Plan-Neofluar oil objective was used. Pre-treated, differentiated HL-60 cells were prepared for Zeiss LSM780 confocal microscopy on fibronectin-coated chambered coverglass as described earlier[66,67]. For global recruitment, the Argon laser was switched on after imaging for 3 min. Photoactivation and image acquisition was done once every 6-7 sec. The laser intensity during image capture was maintained at 0.14–0.17 W/cm$^2$ at the objective, which ensured effective Aldolase A recruitment over the cell periphery without inducing photo damage.

## Cell migration quantification and morphological analysis

All migration track, cell speed or area, and aspect ratio analyses were carried out by segmenting MCF10A or differentiated HL-60 cells on Fiji/ImageJ 1.52i software[73], as described previously[14].

## Wave quantification

We quantified only the waves that originated and propagated within the internal region of the cell's basal surface. Protrusion events observed at the cell edge were not considered traveling waves and were therefore excluded from our wave quantification. The duration of a wave was defined as the time from its initial appearance to its disappearance. If a wave split into multiple parts, the duration was measured until the longest-lasting portion disappeared. In cases where a wave merged with another, its duration was calculated from its origin to the disappearance of the merged wave. The wave length was determined as the maximum lateral span of a wave during its lifetime. The band width corresponded to the width of the line scan shown in Fig. 2l and Supplementary Fig. 1e–h. The velocity was calculated as the average velocity of all segments of a given wave.

### Quantification of biosensors for ATP (Ruby-iATPSnFR), pyruvate (mRuby-PyronicSF), and NADH/NAD + (mCherry-Peredox)

Two channels in each biosensor were captured simultaneously in a confocal microscope. The mask of image was firstly obtained by binarizing the images of mRuby or mCherry channel following despeckling, proper thresholding, and holes-filling in Fiji/ImageJ. The background-removal images were generated by multiplying the images of all channels to the corresponding masks. The Ruby channel of iATP and Pyronic or mCherry of Peredox were added for a very small value to make the denominator non-zero. The ratio images were gained by dividing the background removed cpGFP channel of iATP, Pyronic, or Peredox channel to its processed corresponding Ruby or mCherry channel.

### Statistics and reproducibility

All experiments were independently repeated at least three times with consistent and reproducible results. For statistical analysis, a minimum of $n = 3$ biological replicates was used. All statistical analyses were performed using GraphPad Prism 9. Data are presented as mean ± SD or SEM, as indicated. P-values were calculated using two-tailed t-tests. Sample sizes were determined empirically based on standard practices in the field, with similar group sizes used for experimental and control conditions. Each micrograph shown in the figures represents a typical example (or image series) from $N \geq 3$ independent experiments. P-values are annotated as follows: n.s. (not significant), $P > 0.05$; $*P \leq 0.05$; $**P \leq 0.01$; $***P \leq 0.001$; and $****P \leq 0.0001$. Additional details regarding statistical parameters and methods can be found in the respective figure legends.

### Reporting summary

Further information on research design is available in the Nature Portfolio Reporting Summary linked to this article.

## Data availability

All raw data and associated statistical analyses are provided with this study. The wave data shown in Fig. 7a were reanalyzed and replotted from our previous publication (PMID: 32877650). All other data supporting the findings of this study are available from the corresponding authors upon request. Source data are provided with this paper.

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

## Acknowledgements

The authors thank all members of the Devreotes laboratory, Douglas Robinson, Pablo Iglesias, Miho Iijima, Halimatu Mohammed and Liang Wang (Janetopoulos Lab), and Hideki Nakamura (Inoue Lab, J.H.U.) for helpful discussions and suggestions. We are especially grateful to Yunlu Li for her generous assistance with data analysis. We thank Takanari Inoue (J.H.U.), Stephen Desiderio (J.H.U.), and the Gerbug Wulf (Harvard) laboratories for their generous provision of plasmids. We thank Orion Weiner (UCSF) and Sean Collins (UC Davis) for providing HL-60 cells and transposon plasmids, respectively. We greatly thank Douglas Robinson (J.H.U.) for the provision of AsPC-1 cell line, and Jun Liu (J.H.U.) for the generous provision of SNU-387, Calu-6, HepG2, MDA-MB-231, and HCT116 cell lines. This work was supported by NIH grant R35 GM118177 (to P.N.D.), AFOSR MURI FA95501610052 (to P.N.D.), DARPA HR0011-16-

C-0139 (to P.N.D.), a Cervical Cancer SPORE P50CA098252 Pilot Project Award (to C.H.H.), R01GM136711 (to C.H.H.), the Sol Goldman Pancreatic Cancer Research Center (to C.H.H.), a Johns Hopkins Discovery Award (to C.H.H.), a W.W. Smith Charitable Trust Award (to C.J.) and an NIH grant S10 OD016374 (to S. Kuo of the JHU Microscope Facility).

## Author contributions

H.Z. and P.N.D. conceived the overall project with input from C.J. and C.H.H. H.Z. designed the experimental plan. H.Z. performed majority of the experiments and data analysis. D.S.P. conducted experiments and data analyses in HL-60 cells (Fig. 5h–k and Supplementary Fig. 8). H.Z., D.S.P., J.B., Y.D., and Y.L. designed, and J.B., Y.D., and Y.L. generated expression constructs. H.Z. prepared the figures and movies. H.Z., C.J., C.H.H., and P.N.D. wrote the manuscript. P.N.D. supervised all aspects of the project. All authors agreed to the final submitted manuscript.

## Competing interests

The authors declare no competing interests.
