## [Transparent Peer Review file · Nature Communications]

Self-Organizing Glycolytic Waves Tune Cellular Metabolic States and Fuel Cancer Progression

Corresponding Author: Professor Peter Devreotes

Version 0:

Reviewer comments:

Reviewer #1

(Remarks to the Author)

By directly visualizing enzymes involved in glycolysis and employing biosensors to measure ATP, pyruvate and NADH/NAD⁺ with single-cell imaging, Zhan et al. challenge the long-standing view that glycolysis, a major ATP-generating pathway, occurs uniformly within the cytosol. Instead, they reveal that these glycolytic enzymes are organized as dynamic waves along the cortex of cancer cells. This finding is both compelling and unexpected—especially considering GAPDH is more commonly recognized as a control protein in western blots! They further demonstrated that alteration of wave dynamics caused corresponding changes in metabolites levels using four methods: EGF and insulin stimulation, Ly and Latrunculin drugs, chemically induced reduction of PI(4,5)P₂, and optogenetically induced recruitment of PFK and aldolase. By examining a panel of cancer cell lines, they found a strong correlation between wave activities and ATP level and further propose that the high energy demand in case cells are linked to the ability of these cells to assemble glycolytic waves, thereby providing a novel explanation for the Warburg effect.

Overall, I think these findings are mind-blowing. Given the ubiquitous presence of actin waves in a wide range of cell types, I expect these findings can be reproduced in other systems and have high impact. Additionally, this discovery of the presence of subcellular self-organized waves in glycolytic activity that are compartmentalized, transient, dynamic, and membrane associated carries profound implications in many diverse fields of research such as metabolism and tissue patterning. For example, the direct visualization of such metabolic pattern at the multicellular level is of great emerging interests in the field of developmental patterning (Cao et al., Nature 2024), but technically reaching subcellular resolution is challenging. The findings in this work thus introduce an unexpected layer of complexity that could have far-reaching impact on our understanding of origin as well as heterogeneity of cellular metabolism that will have tremendous impact on tissue patterning in the future. I would like to congratulate the authors for making an important discovery and strongly support the publication of this work, with a few comments below for consideration.

Major issues:

The glycolytic waves themselves have not yet been carefully quantified or fully characterized. Cortical waves have been observed in various experimental systems, each differing in quantitative features. While I understand that exhaustive characterization is not expected in a single study, some basic quantifications would be helpful for future research to better understand the similarities and differences between these waves—particularly since this work has the potential to be a landmark study.

1. It will be good include some information such as speed, duration, and interpeak intervals in the text. From the figures, these waves appear to last a few minutes, with intervals around 20–30 minutes (as a kymograph lasting 2.6 hours shows 5–6 waves), but no quantitative analysis is provided.
2. Among the enzymes and actin involved in the waves, is there any difference in their timing of recruitment?
3. Line 110: PFK shows the strongest enrichment. How was this conclusion reached? The manuscript does not quantify or compare the enrichment levels of each enzyme in a directly comparable manner.
4. Did overexpression of the enzymes affect actin waves?
5. Would glucose starvation inhibit either the glycolysis waves or actin waves?

Given that most methods used to alter glycolytic waves also activate or inhibit signaling pathways, one might question whether the resulting changes in cell migration are due to shifts in signaling or in metabolism. In this model, signaling and metabolism are closely intertwined, making it essential to include controls that can elevate cytosolic ATP without altering the

cortical waves. Achieving this separation may be challenging and will require further insights into the feedback mechanisms involved, but it is crucial for understanding the potential advantage of localizing glycolysis to the plasma membrane in such coordinated manner rather than in the cytosol. Would overexpressing these enzymes in the cytosol alone lead to measurable metabolic changes? Alternatively, is catalytic domain involved in the wave? Would a dead enzyme also participate in the waves without elevating ATP?

The waves of ATP, pyruvate, and NADH are not observed, likely because of small sizes of these molecules and fast diffusion. This is intriguing because if ultimately ATP is not local, what is the purpose of the enrichment of enzymes. On the other hand, NADH waves were reported in neutrophils two decades ago (mysteriously, these papers were subsequently retracted) (Petty et al., PNAS 2001; Kindzelskii et al., PNAS 2002). I am wondering whether one could revisit this with better sensors but this could be beyond the scope of this study.

Minor issue:

1. What is the time resolution of metabolic sensors? Is the decay of ATP significantly slower relative to disappearance of waves or they have similar rates?
2. Number of experiments/cells/statistics have not been provided for all figures.
3. Line 102: half-width of the aldolase waves was approximately twice that of the F-actin waves. Sig. S1F is not clear to demonstrate this. Could the authors provide statistics?
4. Line 197, it is not established that "recruitment of a single enzyme to the plasma membrane enhance overall glycolysis". The data of Fig. 4 only show enhanced actin.
5. Fig. S2 showed that DB reduced ATP level. Would drugs DB and OAR affect waves of enzymes?
6. In many statements in the text such as "The ratio of cpGFP to mRuby decreased ~70% in less than 10 min after addition of DB (Fig. S2D)." "When OAR was applied first, the ratio of cpGFP to mRuby was reduced by less than 10%, while further application of DB reduced the ratio by ~60% (Fig. S2E)." please include standard deviations and sample sizes.
7. The de novo waves in Fig. 4E-G and Video S14 are not convincing. The new clusters after rapamycin did not propagate (Fig. 4F). This could mean that the disassociation is important for wave propagation. Besides, a more upstream enzyme (HK) would be better, if possible.
8. It will be good to provide images corresponding to Fig. 6D (increased waves), Fig 6F (different ATP, in the format of Fig S2C). For each cell types, these values seem to have little variability. Please include sample sizes.
9. Line 324: how was 35% calculated?

Cao, D. et al. Selective utilization of glucose metabolism guides mammalian gastrulation. *Nature* 1–10 (2024)
doi:10.1038/s41586-024-08044-1.

Petty, H. R. & Kindzelskii, A. L. Dissipative metabolic patterns respond during neutrophil transmembrane signaling. *Proc Natl Acad Sci USA* 98, 3145–3149 (2001).

Kindzelskii, A. L. & Petty, H. R. Apparent role of traveling metabolic waves in oxidant release by living neutrophils. *Proc Natl Acad Sci USA* 99, 9207–9212 (2002).

Reviewer #2

(Remarks to the Author)

The authors present data suggesting a wave of glycolysis starting at the plasma membrane following transfection of GFP labelled glycolytic enzymes in the paper entitled "Self-organizing glycolytic waves in plasma membrane fuel cell dynamics and cancer progression". The authors transfect in GFP-labelled glycolytic enzymes (Hexokinase, phosphofructokinase, aldolase, pyruvate kinase, enolase, GAPDH) and investigate cellular localisation using LifeAct-iRFP as an F-actin marker. They show that the GFP-tagged transfected enzymes are localised to sites of F-Actin in waves. They nicely show that disruption of F-actin or glycolysis can alter the localisation through a series of inhibitor studies. Lastly, they confirm that high glycolysis is associated with a high wave activity in a panel of tumour cell lines.

Comments:

- 1: Is the amount of different glycolytic enzymes transfected equivalent? Is it a similar amount per enzyme as PK looks very diffuse across the entire cell, while enolase looks more punctate.
2. Could some of the staining be confirmed with the endogenous proteins to ensure it is not an artefact of overexpression, by co-staining for aldolase and F-actin for example in fixed cells. While the temporal effect will be lost the co-localisation could be assessed.
3. Some higher-resolution imaging (STED/STORM) would be useful to determine the presence of the enzymes at plasma membranes and not merely the leading edges of the cytosol. As the title of the paper may be slightly misleading, as the

plasma membrane association was not confirmed.

4. The addition of EFG and Insulin is known to increase glycolysis so uncoupling that from the effect on F-actin is difficult (e.g PMID: 304448)

Minor comments:

1. The overlay of the GFP-tagged enzymes with LifeAct-iRFP is for a very small region identified by the white arrow in Figure 2. Should the overlays be for a line across the entire cell to show regions of co-localisation and regions of no co-localisation?

2. The LifeAct-iRFP staining looks different in the enolase-GFP cells (video S7). It seems much more diffuse and not localised to leading edges, as was evident in the other videos.

3. Should the Lyn-FRB be data be graphed in Figure 5.

4. Using the Seahorse Extracellular acidification and oxygen consumption assays to confirm the reliance on glycolysis in the cancer cell lines with higher wave activity may also prove useful.

Reviewer #3

(Remarks to the Author)

Reviewer #4

(Remarks to the Author)

In this manuscript, Zhan, H. et al. made an interesting observation that glycolytic enzymes localize to the plasma membrane/cortex of cells and exhibit wave-like behavior. These glycolytic waves co-localize with the well-studied actin waves, suggesting its functional role in generating ATP for cell migration. Applying chemical inhibitors that suppress glycolysis, OXPPOS, and actin polymerization resulted in corresponding changes in the glycolytic wave activities, indicating the interdependence between glycolytic waves, glycolytic activity, and the cytoskeleton. By employing optogenetics, they found that the recruitment of one glycolytic enzyme can lead to the recruitment of other glycolytic enzymes, suggesting their complex formation. Furthermore, they showed that wave activity correlates with metastatic index and cancer cells' dependency on glycolysis, leading the authors to suggest an important role for these glycolytic waves in cancer progression.

Previous studies relying on biochemical approaches found that a number of glycolytic enzymes localize to the cell membrane, forming complexes called glycolytic enzyme metabolon (Puchulu-Campanella, E. et al., 2012, J Biol Chem, Puchulu-Campanella, E. et al., 2013, J Biol Chem, Wang, H. et al., 2024, Nature Metabolism). By applying real-time imaging, this study revealed the phenomenon of the dynamic glycolytic waves in the plasma membrane/cortex— a novel discovery. The authors further developed clever experimental means of controlling the localization of glycolytic enzymes using optogenetics. They extend these findings by exploring the functional role of these glycolytic waves in the context of cancer metabolism. While these findings could offer valuable insights into cancer biology, the evidence supporting the role of these glycolytic waves in cancer progression were correlative, rather than causal. Moreover, there are missing controls in their experiment, questionable data and key unanswered questions. The authors are encouraged to improve the rigor of their work and to tone down their claims. Here, we list several critical points for the authors to address before the study can be considered for publication.

Major points:

1. In Video S9, the cell overexpressing PFK-GFP stimulated with EGF + insulin is the exact SAME cell that was NOT stimulated with EGF + insulin shown in Video S5. The data was duplicated. This warrants an explanation from the authors.

2. The authors make strong claims about the role of glycolytic waves in regulating glycolytic activity and cancer progression, but without sufficient supporting evidence. While their results show strong correlation between glycolytic wave activities, ATP production, and metastatic potential, they do not imply causality. This is elaborated in points 15 & 19. I would suggest the authors to tone down their statements and revise the abstract, title, and the section titles.

3. The major findings of this study are based on the observation that glycolytic enzymes propagate as waves in the cell cortex. These glycolytic waves were observed by introducing GFP-tagged glycolytic enzymes into MCF-10A. However, whether endogenous glycolytic enzymes behave similarly with the exogenous GFP-tagged glycolytic enzymes was not validated. Fluorescent tagging and expression level can affect the localization and function of exogenously-introduced proteins. Since this is the main experimental system of the study, this validation will be important to establish the foundation of their claims. An immunofluorescence co-staining of actin and all the selected glycolytic enzymes in non-engineered cells needs to be performed to confirm their co-localization.

4. There are inconsistencies in the features of the Aldolase-GFP waves among different repeats. For example, the Aldolase-GFP waves in Video S3 have higher signal intensities and greater wavefront widths compared to the waves in Video S1. Can the authors comment on this difference? Can the expression level of Aldolase-GFP affect the wave features?

5. It was not mentioned how general the observation of glycolytic waves is across different individual cells. In cells overexpressing Aldolase-GFP, and other glycolytic enzymes, can the authors provide quantifications on the percentage of cells exhibiting these glycolytic waves? In addition, in cells exhibiting glycolytic waves, are there variations in their wave activities? Perhaps the authors can quantify the wave features, such as velocity, width, and duration of the glycolytic waves, similar to how they characterized the actin wave in their previous study (Zhan, H. et al. 2020, *Developmental Cell*).

6. Kymographs in Figure 1B, 1D, 1F, were presented to help the readers visualize wave propagation shown in the time-lapse images in Figure 1A, 1C, 1E. However, it is unclear how the images shown are linked to the kymographs presented due to the following reasons:

- a. The time frames of the time-lapse images are different from the range of time frames included in the kymographs.
- b. The wave-like behavior in the time-lapse images in Fig. 1A is unconvincing due to the limited time frames presented, and the very small size of the waves. Perhaps the author can show zoomed-in time-lapse images focused on the wave. Again, similar to how they showed it in their previous study (Zhan, H. et al. 2020, *Developmental Cell*).
- c. The kymographs shown did not really highlight the spread of a signal. The authors can try to apply the kymograph analysis in a smaller area focused on the wave. Such strategy has been applied in a previous paper that characterized the propagation of actin waves (Case, L., & Waterman, C. 2011, *PLoS ONE*).

7. The authors cited previous studies that show EGF + insulin treatment can increase actin wave activity, but did not show them in this work. I suggest the authors to perform simultaneous imaging of the Aldolase-GFP wave and actin wave after EGF + insulin treatment in Figure 3A, and quantify their corresponding wave activities.

8. It is unclear how the authors calculated for the “wave activity” in Figure 3B and 3C. The figure legends indicate that the wave activity is “defined as areas of pixels with above threshold aldolase-GFP intensity”. I suggest showing the time-lapse of these thresholded images. At least one representative time-lapse of the thresholded image, aldolase-GFP, and actin will be helpful for the readers to understand Figure 3B.

9. Figure 3B seems problematic. There are some abrupt changes in wave activity, such as the sudden drop followed by sudden recovery of wave activity in cells #8 and #9, and the lack of wave activity in the last time points for cells #1 and #2. Can the authors explain these data points? It will be helpful if the authors showed the movies for all the cells that were quantified here.

10. In Figure 3I-L, LatA was applied to block “the cortical association of the glycolytic enzymes” (lines 151-152). However, the authors did not show any data showing that the glycolytic waves were abolished. In addition, I suggest the authors to show the effect of LatA treatment on the localization of endogenous actin and endogenous glycolytic enzymes using immunofluorescence in non-engineered cells to validate the effect of LatA treatment.

11. LatA treatment can induce cell death by disrupting the actin cytoskeleton network. It will be important to show that the specific dose of LatA used does not affect the cell viability to be able to attribute the decreased ATP levels with the decreased glycolytic enzyme recruitment to the plasma membrane.

12. The authors showed that LY294002 treatment results in decreased ATP levels and cortical wave activity (Figure 3P-R). It is important to also show if LY294002 affects the glycolytic waves.

13. EGF+ insulin stimulation increases glycolytic wave activity (Fig. 3B). In lines 148-150, the authors claim that these results suggest that the increase in wave activity increases glycolytic ATP production. This claim is not supported by the result in Figure 3B. Insulin and EGF treatments have already been shown to increase glycolytic ATP production through various mechanisms (e.g., increase in glucose uptake). Therefore, the increase in ATP after insulin + EGF treatment is not necessarily linked to the increase in wave activity. The authors need to clarify this point and tone-down their claim.

14. The authors wrote that lowering plasma membrane recruitment of PIP2 through the CID system can initiate cortical waves, but no data of cortical waves was shown. Imaging both aldolase-GFP wave (Fig. 4B) and cortical waves simultaneously is suggested.

15. The authors used the CID system to recruit PFK to the plasma membrane. In lines 179-180, they claim that PFK recruitment triggered actin waves. However, the triggered actin waves are not convincing, as they resemble actin aggregation instead. In addition, based on the kymograph in Figure 4F, it seems that the LifeAct-iRFP increased its signal intensity first, before the GFP-FKBP-PFK signal intensity increased. Can the authors comment on this?

16. It is known that light exposure per se can lead to higher cell mobility. As a control for Figure 4H-K, it will be important to perform the same experiments and quantification of cell mobility in non-engineered HL-60 cells.

17. In lines 204-206, the authors claim that the recruitment of glycolytic enzymes to the plasma membrane enhance “ATP production from glycolysis” but there is no sufficient data to support this claim. So far, the increase in glycolytic wave activity and the increase in ATP are correlative, and may not necessarily be causal. For example, in the experiment where

recruitment of PFK to the plasma membrane by CID was performed, the authors did not measure if there are any changes in glycolytic activity.

18. The correlation between ATP levels measured by the ATP biosensor and wave activity in different cell lines is striking (Fig. 6F). How have the authors made sure that the ATP biosensor fluorescence intensity is not dependent on different expression levels of the biosensor in different cells? Can the authors validate this finding using conventional assays for measuring ATP (e.g., colorimetric assays)?

Minor points:

1. The authors interchangeably use actin waves, cytoskeletal waves and cortical waves. To avoid confusion, the authors can choose a single term to use consistently.
2. For the readers' convenience, the unit of the time frames (e.g. min/s/hr) can be placed on the images. The time frames labelled on the images are also suggested to match the time frames labelled on the Supplementary Videos. For example, 1 minute on the time-lapse image is the 1 minute on the Video.
3. For accuracy, in Video S9, "Aldolase" should instead be "Aldolase-GFP". Please apply this correction where applicable.
4. There seems to be a big variation in wave activities after EGF + insulin stimulation in Figure 3B and 3E. To aid readers' visualization and understanding of this variation, the authors can perform hierarchical clustering of the time course data.
5. In Video S10, after DB and OAR was added, the cell seems to drastically change its shape and its protrusions disappear. It is important for the authors to show that the DB and OAR treatment does not affect the cell viability to be able to correctly interpret these results.
6. It is suggested to show the movies for Figure 3G, 3I, and 6F-H.
7. Aggregation (but not localized in moving waves) of overexpressed Aldolase-GFP are observed (central dot-like signals in the cell in Video S1), with intensities higher than the cytosolic signal intensity. Can the authors comment on this?
8. The method by which the authors classified MCF10A M3 and M4 as cells with "higher metastatic index" (lines 215-217) was not indicated.
9. The writing of the manuscript contains multiple instances of sentences with multiple modifying- clauses, run-on sentences, and some grammatical errors. For example, line 59 is a sentence with multiple modifying clauses, "We wondered whether glycolysis can be localized directly to the plasma membrane to accelerate local ATP production to meet the high demand for spatiotemporal regulation of energy at the cell periphery." Readability of the manuscript can be greatly improved if the text can be re-edited.
10. Statistical analyses described in methods does not apply to all figures. For example, Figure 6 and 7 do not use a parametric t-tests. Moreover, p-values should be calculated from a two-tailed t-test instead of "two-tailed p-values".

Version 1:

Reviewer comments:

Reviewer #1

(Remarks to the Author)

The authors have addressed all my concerns. I support its publication.

Reviewer #2

(Remarks to the Author)

The authors have sufficiently addressed the concerns that were raised in the original review.

Reviewer #3

(Remarks to the Author)

Reviewer #4

(Remarks to the Author)

The authors have addressed most of my questions.

We are grateful to the reviewers for their insightful and constructive feedback, which has significantly strengthened our study.

In this revised version, we have thoroughly addressed all concerns raised by the four reviewers. Specifically, we have done the following:

New Experiments

- Most importantly, we validated our findings using immunofluorescence staining of endogenous glycolytic enzymes to confirm their localization in waves, addressing concerns about potential overexpression artifacts (**new Figure 3** and **supplemental Figure S3**).
- Performed luciferase ATP assays to validate ATP biosensor measurements, further strengthening the link between wave activity and metabolic output (**new supplemental Figure S13**).
- Demonstrated that glycolytic enzyme localization in waves is independent of enzymatic activity using catalytically inactive aldolase (**new supplemental Figure S10**).

New Quantifications, Figure & Video Updates, and Rewriting

- Performed additional quantifications and correlation analysis of glycolytic and actin waves, including wave length (maximum lateral expansion), band width, velocity, and duration, for a more comprehensive characterization (**new supplemental Figure S2**).
- Improved the clarity of our kymograph analyses and included zoomed-in views to enhance the visualization of wave dynamics (**Figure S1A-B**).
- Added a new video showing thresholded images of glycolytic waves in a cell before and after EGF and insulin stimulation to better illustrate the quantification strategy (combined with **Video S9**).
- Added a new video demonstrating the coordinated changes in glycolytic and actin waves before and after inhibitor treatments, highlighting their coupled behavior under both basal and inhibited conditions (combined with **Video S11**).
- Revised the manuscript's *title, abstract, subtitles, and conclusions* to clarify the distinction between correlation and causation, ensuring a balanced interpretation of our findings.

All major changes in the main text, figure legends, and video captions are highlighted in red.

REVIEWER COMMENTS

Reviewer #1 (Remarks to the Author):

By directly visualizing enzymes involved in glycolysis and employing biosensors to measure ATP, pyruvate and NADH/NAD⁺ with single-cell imaging, Zhan et al. challenge the long-standing view that glycolysis, a major ATP-generating pathway, occurs uniformly within the cytosol. Instead, they reveal that these glycolytic enzymes are organized as dynamic waves along the cortex of cancer cells. This finding is both compelling and unexpected—especially considering GAPDH is more commonly recognized as a control protein in western blots! They further demonstrated that alteration of wave dynamics caused corresponding changes in metabolites levels using four methods: EGF and insulin stimulation, Ly and Latrunculin drugs, chemically induced reduction of PI(4,5)P₂, and optogenetically induced recruitment of PFK and aldolase. By examining a panel of cancer cell lines, they found a strong correlation between wave activities and ATP level and further propose that the high energy demand in case cells are linked to the ability of these cells to assemble glycolytic waves, thereby providing a novel explanation for the Warburg effect.

Overall, I think these findings are mind-blowing. Given the ubiquitous presence of actin waves in a wide range of cell types, I expect these findings can be reproduced in other systems and have high impact. Additionally, this discovery of the presence of subcellular self-organized waves in glycolytic activity that are compartmentalized, transient, dynamic, and membrane associated carries profound implications in many diverse fields of research such as metabolism and tissue patterning. For example, the direct visualization of such metabolic pattern at the multicellular level is of great emerging interests in the field of developmental patterning (Cao et al., Nature 2024), but technically reaching subcellular resolution is challenging. The findings in this work thus introduce an unexpected layer of complexity that could have far-reaching impact on our understanding of origin as well as heterogeneity of cellular metabolism that will have tremendous impact on tissue patterning in the future. I would like to congratulate the authors for making an important discovery and strongly support the publication of this work, with a few comments below for consideration.

Thank you for the very positive comments.

Major issues:

The glycolytic waves themselves have not yet been carefully quantified or fully characterized. Cortical waves have been observed in various experimental systems, each differing in quantitative features. While I understand that exhaustive characterization is not expected in a single study, some basic quantifications would be helpful for future research to better understand the similarities and differences between these waves—particularly since this work has the potential to be a landmark study.

1. It will be good include some information such as speed, duration, and interpeak intervals in the text. From the figures, these waves appear to last a few minutes, with intervals around 20–30 minutes (as a kymograph lasting 2.6 hours shows 5–6 waves), but no quantitative analysis is provided.

We previously quantified the wavelength (maximum lateral expansion), velocity, and duration of actin waves in a prior study (PMID: 32877650). In this revision, we have newly quantified the wave length (maximum lateral expansion), band width, velocity, and duration for both Aldolase-GFP and LifeAct-RFP expressed in the same cells (**Fig. S2A-D**). As the reviewer noted, wave intervals appear to be around 20–30 minutes based on the kymographs. Our new quantification

confirms that the average wave duration is approximately 23 minutes. While the wave length, velocity, and duration are highly consistent between glycolytic and actin waves, the average band width of aldolase waves is approximately 25% broader than that of actin waves, indicating a slightly more diffusive nature. To better illustrate the coordinated movement of actin and glycolytic waves, we have included correlation plots in this revision as well (**Fig. S2E-H**).

2. Among the enzymes and actin involved in the waves, is there any difference in their timing of recruitment?

As we showed in the figures, all enzymes are more or less colocalized with LifeAct labeled actin (Fig. 2L). There may be slight differences between the different enzymes and actin, but we do not have enough information to make a conclusion.

3. Line 110: PFK shows the strongest enrichment. How was this conclusion reached? The manuscript does not quantify or compare the enrichment levels of each enzyme in a directly comparable manner.

The reviewer is correct; we removed this statement.

4. Did overexpression of the enzymes affect actin waves?

In response to comment 1, we quantified the actin waves in cells overexpressing aldolase (**Fig. S2**). The maximum lateral expansion, velocity, and duration of actin waves in these cells exhibited variability; however, the overall ranges and averages were very similar to those of actin waves in cells without overexpression of glycolytic enzymes, as we previously reported (PMID: 32877650).

5. Would glucose starvation inhibit either the glycolysis waves or actin waves?

Although we did not carry out glucose starvation, the addition of 2-DDG, which competes the binding of glucose to hexokinase, inhibited both the glycolytic and the actin waves.

Given that most methods used to alter glycolytic waves also activate or inhibit signaling pathways, one might question whether the resulting changes in cell migration are due to shifts in signaling or in metabolism. In this model, signaling and metabolism are closely intertwined, making it essential to include controls that can elevate cytosolic ATP without altering the cortical waves. Achieving this separation may be challenging and will require further insights into the feedback mechanisms involved, but it is crucial for understanding the potential advantage of localizing glycolysis to the plasma membrane in such coordinated manner rather than in the cytosol. Would overexpressing these enzymes in the cytosol alone lead to measurable metabolic changes? Alternatively, is catalytic domain involved in the wave? Would a dead enzyme also participate in the waves without elevating ATP?

We agree that this separation is important but challenging in experimental design. We are not sure how to express enzymes that will remain only in the cytosol. As a partial answer to your query, we produced a catalytically dead aldolase (D34S) and expressed it. It appears in the waves similarly to the WT enzyme. This data was added (**Fig. S10**).

The waves of ATP, pyruvate, and NADH are not observed, likely because of small sizes of these molecules

and fast diffusion. This is intriguing because if ultimately ATP is not local, what is the purpose of the enrichment of enzymes. On the other hand, NADH waves were reported in neutrophils two decades ago (mysteriously, these papers were subsequently retracted) (Petty et al., PNAS 2001; Kindzelskii et al., PNAS 2002). I am wondering whether one could revisit this with better sensors but this could be beyond the scope of this study.

We attempted this using a membrane-tethered version of our ATP sensor. The results suggested elevated signals at the cell edges, but it remains unclear whether these signals are associated with waves. However, further studies and improved sensors are needed to draw a definitive conclusion. It remains an exciting idea for a follow-up study.

Minor issue:

1. What is the time resolution of metabolic sensors? Is the decay of ATP significantly slower relative to disappearance of waves or they have similar rates?

Comparison of the responses to DB (glycolysis inhibitors) showed that the rate of ATP disappearance (**Fig. S4D**) is similar to that of wave disappearance (**Fig. S9B**).

2. Number of experiments/cells/statistics have not been provided for all figures.

We have provided the number of all cells and statistical details for all relevant panels in the corresponding figure legends. Regarding the number of experiments, some are specified in the figure legends for individual panels, while the rest are indicated at the end of **Fig. 4**: “All the cells plotted and quantified in each panel were from at least three independent experiments, which is consistent throughout the later figures.”

3. Line 102: half-width of the aldolase waves was approximately twice that of the F-actin waves. Fig. S1F is not clear to demonstrate this. Could the authors provide statistics?

We measured the band width of 20 coupled aldolase and actin waves (new data included in **Fig. S2B**), and found that the aldolase waves were slightly (~25%) wider than the F-actin waves. We have updated the main text accordingly. Scans of the band widths of the other enzymes versus actin waves are provided in **Fig. 2L**.

4. Line 197, it is not established that “recruitment of a single enzyme to the plasma membrane enhance overall glycolysis”. The data of Fig. 4 only show enhanced actin.

We have rephrased and corrected this sentence.

5. Fig. S2 showed that DB reduced ATP level. Would drugs DB and OAR affect waves of enzymes?

As shown in **Fig. S9B** and **S9I**, DB dramatically reduced the actin (LifeAct-labeled) waves, while OAR had minimal effect on them. Given the strong coupling behavior between waves of actin and glycolytic enzymes (**Fig. S2**), DB and OAR are likely to affect enzyme waves in a similar pattern.

6. In many statements in the text such as “The ratio of cpGFP to mRuby decreased ~70% in less than 10

min after addition of DB (Fig. S2D). “ “When OAR was applied first, the ratio of cpGFP to mRuby was reduced by less than 10%, while further application of DB reduced the ratio by ~60% (Fig. S2E).” please include standard deviations and sample sizes.

Statistics such as Standard Error of the Mean and sample sizes are provided in corresponding legends of **Fig. S4D, E**.

7. *The de novo waves in Fig. 4E-G and Video S14 are not convincing. The new clusters after rapamycin did not propagate (Fig. 4F). This could mean that the disassociation is important for wave propagation. Besides, a more upstream enzyme (HK) would be better, if possible.*

We have rephrased our description. There is clearly an increase in actin polymerization, but as the reviewer points out, it is more like star-shaped patches than propagating waves.

8. *It will be good to provide images corresponding to Fig. 6D (increased waves), Fig 6F (different ATP, in the format of Fig S2C). For each cell types, these values seem to have little variability. Please include sample sizes.*

We had added images showing increases in wave activity in various cell lines (**Fig. S12**). Images for the ATP sensor did not show any subcellular structures, and we quantified the signals of the entire cells. We do not think adding these images will provide further information, but will be happy to provide the original images if necessary. The sample sizes were initially included in the figure legend of our manuscript.

9. *Line 324: how was 35% calculated?*

Newly added luciferase assays indicate that DB essentially eliminates all ATP (**Fig. S13**), but the ratio signal (cpGFP/mRuby) of the ATP biosensor dropped by 70% (**Fig. S4D**). In these cells, the biosensor signals reduced by 25% with LY treatment (**Fig. 4P**). Assuming that the sensor signal is linear within the ATP range of the experiment, 25%/70% is 35.7%. To better address this, we also updated the corresponding sentence in the discussion: “*Our findings suggest that ATP production from these glycolytic waves may constitute approximately 35% of the total ATP generated through glycolysis, given that iATP cpGFP/mRuby levels were reduced by ~25% with LY or LatA treatment and ~70% with high DB.*”

Cao, D. et al. Selective utilization of glucose metabolism guides mammalian gastrulation. Nature 1–10 (2024) doi:10.1038/s41586-024-08044-1.

Petty, H. R. & Kindzelskii, A. L. Dissipative metabolic patterns respond during neutrophil transmembrane signaling. Proc Natl Acad Sci USA 98, 3145–3149 (2001).

Kindzelskii, A. L. & Petty, H. R. Apparent role of traveling metabolic waves in oxidant release by living neutrophils. Proc Natl Acad Sci USA 99, 9207–9212 (2002).

Reviewer #2 (Remarks to the Author):

The authors present data suggesting a wave of glycolysis starting at the plasma membrane following transfection of GFP labelled glycolytic enzymes in the paper entitled ‘Self-organizing glycolytic waves in plasma membrane fuel cell dynamics and cancer progression’. The authors transfect in GFP-labelled glycolytic enzymes (Hexokinase, phosphofructokinase, aldolase, pyruvate kinase, enolase, GAPDH) and investigate cellular localisation using LifeAct-iRFP as an F-actin marker. They show that the GFP-tagged transfected enzymes are localised to sites of F-Actin in waves. They nicely show that disruption of F-actin or glycolysis can alter the localisation through a series of inhibitor studies. Lastly, they confirm that high glycolysis is associated with a high wave activity in a panel of tumour cell lines.

Comments:

1: Is the amount of different glycolytic enzymes transfected equivalent? Is it a similar amount per enzyme as PK looks very diffuse across the entire cell, while enolase looks more punctate.

Yes, the amounts of the different glycolytic enzymes transfected were similar, although heterogeneous, as determined by fluorescence intensity under consistent microscopy laser settings.

Additionally, we examined the endogenous levels of these enzymes, as requested in comment 2 (**Fig. 3**).

2. Could some of the staining be confirmed with the endogenous proteins to ensure it is not an artefact of overexpression, by co-staining for aldolase and F-actin for example in fixed cells. While the temporal effect will be lost the co-localisation could be assessed.

We thank the reviewer for this constructive comment. To address this, we first demonstrated that PFK-GFP waves and LifeAct-RFP-labeled actin waves can be retained after fixation by adding a fixation reagent (4% PFA) during video recording (**Fig. 3A** and **S3**). We then performed immunostaining for endogenous enzymes using antibodies against aldolase, PFK, GAPDH, enolase, and hexokinase in fixed cells expressing LifeAct. The observed waves of these endogenous enzymes were highly consistent with the results obtained from plasmid expression (**Fig. 3B-K**). We have included a **new Fig. 3** and supplemental **Fig. S3** to illustrate these findings.

3. Some higher-resolution imaging (STED/STORM) would be useful to determine the presence of the enzymes at plasma membranes and not merely the leading edges of the cytosol. As the title of the paper may be slightly misleading, as the plasma membrane association was not confirmed.

The reviewer is correct. We are uncertain whether the waves of glycolytic enzymes are associated with the plasma membrane, the cellular cortex, or both. Therefore, we have revised the title, abstract, and descriptions in the Results and Discussion sections accordingly.

4. The addition of EFG and Insulin is known to increase glycolysis so uncoupling that from the effect on F-actin is difficult (e.g PMID: 304448)

We agree that disentangling the two possible mechanisms is challenging, as EGF/Insulin stimulation enhances both waves and PIP3 levels (PI3K activity). We believe the reviewer is referring to the role of PI3K signaling in this process. PI3K can activate Akt, which phosphorylates and stimulates various glycolytic enzymes. We have demonstrated that the actin inhibitor Latrunculin A blocks both waves and ATP production in response to EGF/Insulin (**Fig. 4G-L**), even though PI3K activity remains elevated under these conditions (**Fig. 4M**). While PI3K signaling can activate downstream glycolytic enzymes, it cannot compensate for the disruption of waves that drive localized glycolysis in this experimental setting.

Minor comments:

1. The overlay of the GFP-tagged enzymes with LifeAct-iRFP is for a very small region identified by the white arrow in Figure 2. Should the overlays be for a line across the entire cell to show regions of co-localisation and regions of no co-localisation?

Our main point in **Figure 2L** was to show that the width of the LifeAct peak is narrower than that of GFP/RFP tagged glycolytic enzymes. A longer line scan across the entire cell to demonstrate the regions of co-localization and regions of no co-localization was shown in **Figure 1F**.

2. The LifeAct-iRFP staining looks different in the enolase-GFP cells (video S7). It seems much more diffuse and not localised to leading edges, as was evident in the other videos.

The LifeAct-iRFP expression and fluorescence intensity is low in this cell, giving that impression.

3. Should the Lyn-FRB be data be graphed in Figure 5.

In the experimental design presented in Figure 5 (now Figure 6), Lyn-FRB was not fused to a fluorescent tag. However, in a similar experiment shown in Figure 5B, Lyn-FRB was tagged with CFP. The CFP-Lyn-FRB and Aldolase-GFP channels are displayed in Figure S7B, where images from different focal planes confirm that Aldolase-GFP is enriched in peripheral waves, whereas CFP-Lyn-FRB is evenly distributed, serving as a marker for the plasma membrane.

4. Using the Seahorse Extracellular acidification and oxygen consumption assays to confirm the reliance on glycolysis in the cancer cell lines with higher wave activity may also prove useful.

We thank the reviewer for this comment. In the revised manuscript, we employed luciferase-based biochemical assays to quantify changes in cellular ATP levels following the inhibition of OXPHOS, glycolysis, and wave activities, as well as stimulation with EGF and insulin. These results are consistent with those obtained using the ATP biosensor. The new data have been included in **Figure S13**.

Reviewer #3 (Remarks to the Author):

Reviewer #4 (Remarks to the Author):

In this manuscript, Zhan, H. et al. made an interesting observation that glycolytic enzymes localize to the plasma membrane/cortex of cells and exhibit wave-like behavior. These glycolytic waves co-localize with the well-studied actin waves, suggesting its functional role in generating ATP for cell migration. Applying chemical inhibitors that suppress glycolysis, OXPHOS, and actin polymerization resulted in corresponding changes in the glycolytic wave activities, indicating the interdependence between glycolytic waves, glycolytic activity, and the cytoskeleton. By employing optogenetics, they found that the recruitment of one glycolytic enzyme can lead to the recruitment of other glycolytic enzymes, suggesting their complex formation. Furthermore, they showed that wave activity correlates with metastatic index and cancer cells' dependency on glycolysis, leading the authors to suggest an important role for these glycolytic waves in cancer progression.

Previous studies relying on biochemical approaches found that a number of glycolytic enzymes localize to the cell membrane, forming complexes called glycolytic enzyme metabolon (Puchulu-Campanella, E. et al., 2012, J Biol Chem, Puchulu-Campanella, E. et al., 2013, J Biol Chem, Wang, H. et al., 2024, Nature Metabolism). By applying real-time imaging, this study revealed the phenomenon of the dynamic glycolytic waves in the plasma membrane/cortex—a novel discovery. The authors further developed clever experimental means of controlling the localization of glycolytic enzymes using optogenetics. They extend these findings by exploring the functional role of these glycolytic waves in the context of cancer metabolism. While these findings could offer valuable insights into cancer biology, the evidence supporting the role of these glycolytic waves in cancer progression were correlative, rather than causal. Moreover, there are missing controls in their experiment, questionable data and key unanswered questions. The authors are encouraged to improve the rigor of their work and to tone down their claims. Here, we list several critical points for the authors to address before the study can be considered for publication.

Major points:

1. In Video S9, the cell overexpressing PFK-GFP stimulated with EGF + insulin is the exact SAME cell that was NOT stimulated with EGF + insulin shown in Video S5. The data was duplicated. This warrants an explanation from the authors.

We used the same cell in Video S5 to illustrate the coordination of PFK waves with F-actin waves in the presence of EGF and insulin, as MCF-10A-M3 cells are typically cultured with these factors. In Video S9, we later demonstrated the effects of EGF and insulin stimulation by including before-and-after conditions in the same cell. That is, we examined the same cell over different time frames. We did not initially consider this to be an issue; however, to clarify the distinction, we have now cropped Video S5 to show only the waves after the addition of EGF and insulin. In the original Video S9, we had included another cell as an additional example of PFK-GFP wave changes upon stimulation. Furthermore, we have added another example of a cell displaying coordinated PFK-GFP and LifeAct-RFP waves in the **new Figs. 3A and S3**.

2. The authors make strong claims about the role of glycolytic waves in regulating glycolytic activity and cancer progression, but without sufficient supporting evidence. While their results show strong correlation between glycolytic wave activities, ATP production, and metastatic potential, they do not

imply causality. This is elaborated in points 15 & 19. I would suggest the authors to tone down their statements and revise the abstract, title, and the section titles.

We have changed the title, abstract, the section titles, and descriptions in the results and discussion sections accordingly.

3. The major findings of this study are based on the observation that glycolytic enzymes propagate as waves in the cell cortex. These glycolytic waves were observed by introducing GFP-tagged glycolytic enzymes into MCF-10A. However, whether endogenous glycolytic enzymes behave similarly with the exogenous GFP-tagged glycolytic enzymes was not validated. Fluorescent tagging and expression level can affect the localization and function of exogenously-introduced proteins. Since this is the main experimental system of the study, this validation will be important to establish the foundation of their claims. An immunofluorescence co-staining of actin and all the selected glycolytic enzymes in non-engineered cells needs to be performed to confirm their co-localization.

To address this, we first demonstrated that PFK-GFP waves and LifeAct-RFP-labeled actin waves can be retained after fixation by adding a fixation reagent (4% PFA) during video recording (**Fig. 3A** and **S3**). We then performed immunostaining for endogenous enzymes using antibodies against aldolase, PFK, GAPDH, enolase, and hexokinase in fixed cells expressing LifeAct. The observed waves of these endogenous enzymes were highly consistent with the results obtained from plasmid expression (**Fig. 3B-K**). We have included a **new Fig. 3** and supplemental **Fig. S3** to illustrate these findings.

We did not use immunofluorescence staining for actin because antibodies against actin do not effectively highlight F-actin waves, particularly the newly formed branched actin wave structures. Instead, we utilized cells expressing LifeAct-RFP to demonstrate the co-localization of endogenous glycolytic enzymes with actin waves. Since LifeAct expression varies among cells, the merged images clearly show that the presence of endogenous glycolytic waves is independent of LifeAct expression levels (**Fig. 3I-J**).

We thank the reviewer for this constructive comment, which has significantly improved the manuscript.

4. There are inconsistencies in the features of the Aldolase-GFP waves among different repeats. For example, the Aldolase-GFP waves in Video S3 have higher signal intensities and greater wavefront widths compared to the waves in Video S1. Can the authors comment on this difference? Can the expression level of Aldolase-GFP affect the wave features?

The waves are self-organized events with inherent variability. In this revision, we have added new quantifications of aldolase wave characteristics, including wave length (maximum lateral expansion), band width, velocity, and duration. As shown in **Fig. S2A-D**, these features exhibit significant variability.

We demonstrated the coupled behaviors of actin and glycolytic waves, through correlation analysis as well (**Fig. S2E-H**). Additionally, we quantified actin waves in cells expressing aldolase at different levels (**Fig. S2**). While the maximum lateral expansion, velocity, and duration of actin waves varied among cells, their overall ranges and averages remained highly similar to those of actin waves in cells without overexpression of glycolytic enzymes in our

previous report (**PMID: 32877650**). Therefore, we do not believe that the expression level of Aldolase-GFP affects wave features.

Moreover, as discussed in **point 3**, endogenous glycolytic enzymes localize to the waves regardless of whether Aldolase-GFP is expressed (**Fig. 3**). Taken together, these findings indicate that wave dynamics are independent of Aldolase-GFP expression levels.

5. It was not mentioned how general the observation of glycolytic waves is across different individual cells. In cells overexpressing Aldolase-GFP, and other glycolytic enzymes, can the authors provide quantifications on the percentage of cells exhibiting these glycolytic waves? In addition, in cells exhibiting glycolytic waves, are there variations in their wave activities? Perhaps the authors can quantify the wave features, such as velocity, width, and duration of the glycolytic waves, similar to how they characterized the actin wave in their previous study (Zhan, H. et al. 2020, Developmental Cell).

For the quantifications on the percentage of cells exhibiting these glycolytic waves, it is already presented in the **original Fig. 6A, 6D (new Fig. 7A, 7D)**. In non-cancer parental control M1 cell lines the percentage of cells exhibiting waves is ~ 10%, and in the cancer cell lines with sequentially enhanced malignancy and metastatic index, the percentages are ~ 30% in M2 cells, ~80% in M3 cells, and ~90% in M4 cells. Across a series of cancer cells derived from different tissues, the percentages vary from ~15% to ~95%.

For variations in wave activities, we have discussed in **point 4** above.

We also did the new quantification on features of glycolytic waves as the reviewer suggested, and we have added as **Fig. S2** in the revised manuscript.

6. Kymographs in Figure 1B, 1D, 1F, were presented to help the readers visualize wave propagation shown in the time-lapse images in Figure 1A, 1C, 1E. However, it is unclear how the images shown are linked to the kymographs presented due to the following reasons:

a. The time frames of the time-lapse images are different from the range of time frames included in the kymographs.

b. The wave-like behavior in the time-lapse images in Fig. 1A is unconvincing due to the limited time frames presented, and the very small size of the waves. Perhaps the author can show zoomed-in time-lapse images focused on the wave. Again, similar to how they showed it in their previous study (Zhan, H. et al. 2020, Developmental Cell).

c. The kymographs shown did not really highlight the spread of a signal. The authors can try to apply the kymograph analysis in a smaller area focused on the wave. Such strategy has been applied in a previous paper that characterized the propagation of actin waves (Case, L., & Waterman, C. 2011, PLoS ONE).

a. For **Figure 1B, 1D, 1F**, we intended to show the clear propagating path of an individual wave, so we showed a time window of several minutes. For the kymographs in **Figure 1A, 1C, 1E**, we intended to show more waves events in a longer time range. This is why the time frames do not match.

b. It is true that the waves appeared small in the images; but we think it is important to show the entire cell in the images. For **Fig.1A**, we have now included a zoomed-in box in **new Fig. S1A** to show the waves in more detail to enhance the readability.

c. We also included a “zoomed-in” box in **new Fig. S1B** for **Fig. 1B**.

7. The authors cited previous studies that show EGF + insulin treatment can increase actin wave activity, but did not show them in this work. I suggest the authors to perform simultaneous imaging of the Aldolase-GFP wave and actin wave after EGF + insulin treatment in Figure 3A, and quantify their corresponding wave activities.

The effects of EGF and insulin on glycolytic waves are shown in **Fig 4**, and the effect on actin waves were shown in our previous published paper (**PMID: 32877650**). The coupled behavior between glycolytic waves and actin waves are quantified in the **new Fig. S2**. Therefore, the changes in actin waves are expected to be similar to the changes in glycolytic waves upon stimulation of growth factors.

8. It is unclear how the authors calculated for the “wave activity” in Figure 3B and 3C. The figure legends indicate that the wave activity is “defined as areas of pixels with above threshold aldolase-GFP intensity”. I suggest showing the time-lapse of these thresholded images. At least one representative time-lapse of the thresholded image, aldolase-GFP, and actin will be helpful for the readers to understand Figure 3B.

Shown below are frames from a video of Aldolase-GFP before and after thresholding, as well as before and after EGF & insulin stimulation. The data presented in **Fig. 3B** (now **new Fig. 4B**) were quantified based on the total intensity of the thresholded images.

We thank the reviewer for the suggestion. For the better understanding of the readers, we have incorporated a thresholded video into the existing **Video S9** to display the thresholded Aldolase-GFP waves alongside the original Aldolase waves of Aldolase_Cell 2.

9. Figure 3B seems problematic. There are some abrupt changes in wave activity, such as the sudden drop followed by sudden recovery of wave activity in cells #8 and #9, and the lack of wave activity in the last time points for cells #1 and #2. Can the authors explain these data points? It will be helpful if the authors showed the movies for all the cells that were quantified here.

During live cell imaging, some frames were occasionally lost due to factors such as definite focus failure. The consecutive frames below, taken from one of the videos in question, illustrate an instance of a failed frame during time-lapse acquisition (original Aldolase-GFP). This loss does not reflect an actual change in wave activity.

10. In Figure 3I-L, LatA was applied to block “the cortical association of the glycolytic enzymes” (lines 151-152). However, the authors did not show any data showing that the glycolytic waves were abolished. In addition, I suggest the authors to show the effect of LatA treatment on the localization of endogenous actin and endogenous glycolytic enzymes using immunofluorescence in non-engineered cells to validate the effect of LatA treatment.

Comparison of the original Fig. 3A and 3M (now **new Fig. 4A and 4M**) shows that aldolase and PFK waves were abolished after LatA treatment. In the **newly added Fig. 3**, included in response to point 3, we demonstrated that endogenous glycolytic enzymes are present in actin waves. The elimination of endogenous actin waves by LatA is a well-established phenomenon in the field.

11. LatA treatment can induce cell death by disrupting the actin cytoskeleton network. It will be important to show that the specific dose of LatA used does not affect the cell viability to be able to attribute the decreased ATP levels with the decreased glycolytic enzyme recruitment to the plasma membrane.

As shown in the original Fig. 3M (now **new Fig. 4M**), LatA treated cells still respond to EGF& Insulin by producing PIP3, indicating that cells are viable. It is typical that this short duration of Lat A treatment does not affect cell viability. Previously, we had washed out the Latrunculin A and returned cells to the incubator and found that they are well attached with normal spreading morphology the next day.

12. The authors showed that LY294002 treatment results in decreased ATP levels and cortical wave activity (Figure 3P-R). It is important to also show if LY294002 affects the glycolytic waves.

We have added one new video to the existing **Video S14** as a typical example to show that LY294002 affects glycolytic waves similar to its effect on actin waves.

13. EGF+ insulin stimulation increases glycolytic wave activity (Fig. 3B). In lines 148-150, the authors claim that these results suggest that the increase in wave activity increases glycolytic ATP production. This claim is not supported by the result in Figure 3B. Insulin and EGF treatments have already been shown to increase glycolytic ATP production through various mechanisms (e.g., increase in glucose uptake). Therefore, the increase in ATP after insulin + EGF treatment is not necessarily linked to the increase in wave activity. The authors need to clarify this point and tone-down their claim.

We change the statement to “This finding suggests that the induced recruitment of the glycolytic enzymes, concentrating them into the waves, may account for the increase in glycolytic activity and ATP production.”

14. The authors wrote that lowering plasma membrane recruitment of PIP2 through the CID system can initiate cortical waves, but no data of cortical waves was shown. Imaging both aldolase-GFP wave (Fig. 4B) and cortical waves simultaneously is suggested.

The increase of cortical waves (actin waves) upon PIP2 reduction was shown in our previous paper (PMID: 32877650). Due to the limitations of colors for microscopy, we were not able to image the aldolase, F-actin, and CID simultaneously. However, in this revision, we have added data showing the tight correlation between F-actin waves and glycolytic waves (Fig. S2).

15. The authors used the CID system to recruit PFK to the plasma membrane. In lines 179-180, they claim that PFK recruitment triggered actin waves. However, the triggered actin waves are not convincing, as they resemble actin aggregation instead. In addition, based on the kymograph in Figure 4F, it seems that the LifeAct-iRFP increased its signal intensity first, before the GFP-FKBP-PFK signal intensity increased. Can the authors comment on this?

We agree. We revised the corresponding description to “*Surprisingly, in MCF-10A M3 cells, recruitment of PFK to the plasma membrane by CID triggered cell spreading and the appearance of dynamic actin patches.*”

For Fig. 4F (now new Fig. 5F), there is some spontaneous activity in LifeAct, as is typical, and it further increases after rapamycin induced recruitment of PFK.

16. It is known that light exposure per se can lead to higher cell mobility. As a control for Figure 4H-K, it will be important to perform the same experiments and quantification of cell mobility in non-engineered HL-60 cells.

As shown in Fig. S8B-C, in cells with no observable aldolase expression, light exposure led to no significant changes in cell migration speed, polarity, and spreading.

17. In lines 204-206, the authors claim that the recruitment of glycolytic enzymes to the plasma membrane enhance “ATP production from glycolysis” but there is no sufficient data to support this claim. So far, the increase in glycolytic wave activity and the increase in ATP are correlative, and may not necessarily be causal. For example, in the experiment where recruitment of PFK to the plasma membrane by CID was performed, the authors did not measure if there are any changes in glycolytic activity.

The reviewer is correct and we thank the reviewer for pointing this out. Therefore, we changed the sentence to “*In either case, the coordinated recruitment of glycolytic enzymes can explain why the recruitment of one glycolytic enzyme is sufficient to induce dynamic actin patches and protrusive activities*”.

18. The correlation between ATP levels measured by the ATP biosensor and wave activity in different cell lines is striking (Fig. 6F). How have the authors made sure that the ATP biosensor fluorescence intensity is not dependent on different expression levels of the biosensor in different cells? Can the authors validate this finding using conventional assays for measuring ATP (e.g., colorimetric assays)?

The ATP biosensor is designed to account for differences in expression level (Fig. S4A). The mRuby is not affected by the ATP binding to the biosensor while the fluorescence intensity of

cpGFP is enhanced upon the binding of ATP. Since mRuby and cpGFP are in the same molecule, the ratio of cpGFP to mRuby is independent of the expression level but correlated with the relative level of ATP.

We have performed ATP luciferase assays to measure the bulk ATP changes in MCF-10A-M3 and MDA-MB-231 cells upon treatment with LatA, LY294002, DB, OAR, and EFG & insulin stimulation (after starvation), and these results are consistent with what we have observed via ATP biosensor and also help calibrate the ATP biosensor (**Fig. S13**).

Minor points:

1. The authors interchangeably use actin waves, cytoskeletal waves and cortical waves. To avoid confusion, the authors can choose a single term to use consistently.

We have tried to stick to “actin waves” and have updated the revised manuscript accordingly.

2. For the readers’ convenience, the unit of the time frames (e.g. min/s/hr) can be placed on the images. The time frames labelled on the images are also suggested to match the time frames labelled on the Supplementary Videos. For example, 1 minute on the time-lapse image is the 1 minute on the Video.

We understand the request to relabel all timestamps in the videos. However, given the scope of the change, this would require a significant amount of time while yielding minimal impact on the overall conclusions. The current timestamps are clear and allow viewers to follow along effectively. If there are specific instances where the labels are unclear or could improve comprehension, we would be happy to address those directly. Please let us know if you have particular concerns.

3. For accuracy, in Video S9, “Aldolase” should instead be “Aldolase-GFP”. Please apply this correction where applicable.

Due to space limitation of the videos, we use Aldolase, but in the video captions (legends) we always indicated the correct nomenclature.

4. There seems to be a big variation in wave activities after EGF + insulin stimulation in Figure 3B and 3E. To aid readers’ visualization and understanding of this variation, the authors can perform hierarchical clustering of the time course data.

We have reordered the traces according to the average activity in **new Fig. 4B and 4E**, as the reviewer suggested.

5. In Video S10, after DB and OAR was added, the cell seems to drastically change its shape and its protrusions disappear. It is important for the authors to show that the DB and OAR treatment does not affect the cell viability to be able to correctly interpret these results.

Specifically, it is after adding DB, that the drastic change happens. We believe this phenotype is significant, as glycolysis is acutely required for cell dynamics, but it does not indicate a loss of viability. We added a sentence to the main text “*The effects of inhibitors were reversible. When*

they were washed out and cells returned to incubator, we observed normal morphology the next day.”

6. It is suggested to show the movies for Figure 3G, 3I, and 6F-H.

Fig. 6F-H (now **new Fig. 7F-H**) are plots of results in Fig. 6D-E (**new Fig. 7D-E**), and videos of representative cells in these seven cell lines showing the wave activities are already included as **Video S17** in the original manuscript. The images in Fig 3G, I (**new Fig. 4G, I**) are intended to show representative time points of a cell responding to the treatment conditions and many cells have already been quantified. Adding the entire videos may not provide much new information.

7. Aggregation (but not localized in moving waves) of overexpressed Aldolase-GFP are observed (central dot-like signals in the cell in Video S1), with intensities higher than the cytosolic signal intensity. Can the authors comment on this?

We had not noticed this but we agree that there are “patches” in this video. This dynamic patch-like structure, which is not travelling with the waves, may be standing waves as we have reported previously in other molecules (PMID: 32821814, PMID: 32877650).

8. The method by which the authors classified MCF10A M3 and M4 as cells with “higher metastatic index” (lines 215-217) was not indicated.

These cell lines have been previously categorized by others (PMID: 8546221, PMID: 11261825, PMID: 23527039). We have added these citations in the revised manuscript.

9. The writing of the manuscript contains multiple instances of sentences with multiple modifying-clauses, run-on sentences, and some grammatical errors. For example, line 59 is a sentence with multiple modifying clauses, “We wondered whether glycolysis can be localized directly to the plasma membrane to accelerate local ATP production to meet the high demand for spatiotemporal regulation of energy at the cell periphery.” Readability of the manuscript can be greatly improved if the text can be re-edited.

We have updated our manuscript accordingly: **“We hypothesized that glycolysis might be localized directly to the plasma membrane, providing a spatial and temporal mechanism to accelerate local ATP production.”**

We also enhanced our writing in other places.

10. Statistical analyses described in methods does not apply to all figures. For example, Figure 6 and 7 do not use a parametric t-tests. Moreover, p-values should be calculated from a two-tailed t-test instead of “two-tailed p-values”.

Thanks for pointing out this. We have updated the statistical analyses description in methods to: **“P-values were calculated using two-tailed t-tests.”**; and **“Further details of statistical parameters and methods are reported in the corresponding figure legends.”**